# The sockeye salmon genome, transcriptome, and analyses identifying population defining regions of the genome

Kris A. Christensen[1,2]*, Eric B. Rondeau[1,2,3], David R. Minkley[1,2], Dionne Sakhrani[1], Carlo A. Biagi[1], Anne-Marie Flores[2], Ruth E. Withler[3], Scott A. Pavey[4], Terry D. Beacham[3], Theresa Godin[5], Eric B. Taylor[6], Michael A. Russello[7], Robert H. Devlin[1], Ben F. Koop[2]*

**1** Fisheries and Oceans Canada, West Vancouver, British Columbia, Canada, **2** University of Victoria, Victoria, British Columbia, Canada, **3** Pacific Biological Station, Fisheries and Oceans Canada, Nanaimo, British Columbia, Canada, **4** University of New Brunswick, Biological Sciences, Saint John, New Brunswick, Canada, **5** Freshwater Fisheries Society of British Columbia, Vancouver, British Columbia, Canada, **6** Department of Zoology, Biodiversity Research Centre and Beaty Biodiversity Museum, University of British Columbia, Vancouver, British Columbia, Canada, **7** University of British Columbia, Okanagan, British Columbia, Canada

* kris.christensen@wsu.edu (KAC); bkoop@uvic.ca (BFK)

**Data Availability Statement:** Raw data has been deposited to the National Center for Biotechnology Information (NCBI) under BioProject PRJNA530256 (https://www.ncbi.nlm.nih.gov/

## Abstract

Sockeye salmon (*Oncorhynchus nerka*) is a commercially and culturally important species to the people that live along the northern Pacific Ocean coast. There are two main sockeye salmon ecotypes—the ocean-going (anadromous) ecotype and the fresh-water ecotype known as kokanee. The goal of this study was to better understand the population structure of sockeye salmon and identify possible genomic differences among populations and between the two ecotypes. In pursuit of this goal, we generated the first reference sockeye salmon genome assembly and an RNA-seq transcriptome data set to better annotate features of the assembly. Resequenced whole-genomes of 140 sockeye salmon and kokanee were analyzed to understand population structure and identify genomic differences between ecotypes. Three distinct geographic and genetic groups were identified from analyses of the resequencing data. Nucleotide variants in an immunoglobulin heavy chain variable gene cluster on chromosome 26 were found to differentiate the northwestern group from the southern and upper Columbia River groups. Several candidate genes were found to be associated with the kokanee ecotype. Many of these genes were related to ammonia tolerance or vision. Finally, the sex chromosomes of this species were better characterized, and an alternative sex-determination mechanism was identified in a subset of upper Columbia River kokanee.

## Introduction

Sockeye salmon (*Oncorhynchus nerka*) are one of eight species of Pacific salmon and trout native to the North Pacific Ocean where they are of tremendous economic and cultural significance. The return of sockeye salmon from the Pacific Ocean to rivers and lakes is part of an

bioproject/PRJNA530256/). Custom scripts and sample information can be found in supplemental files.

**Funding:** Funding for this study was provided by Fisheries and Oceans Canada (https://www.dfo-mpo.gc.ca/index-eng.htm) under the Canadian Regulatory System for Biotechnology to RHD. BFK's, MAR's, and EBT's research is supported by the Natural Sciences and Engineering Research Council of Canada (https://www.nserc-crsng.gc.ca/index_eng.asp) (EBT - Discovery and Equipment grants programs). The funders had no role in study design, data collection and analysis, decision to publish, or preparation of the manuscript.

**Competing interests:** The authors have declared that no competing interests exist.

ancient series of migrations that began with the emergence of the species several million years ago (reviewed in [1–5]). Spawning sockeye salmon, with their bright red bodies, pulse at various times during summer and fall through streams from the Columbia River to the Mackenzie River (Northwest Territories, Canada) in North America, and from Hokkaido, Japan to the Chukotka Peninsula in Asia [6]. The largest concentrations of sockeye salmon, and where most commercial catches are taken, centre around Bristol Bay (Alaska, USA), the Fraser River (BC, Canada), and the Kamchatka Peninsula (Kamchatka, Russia) [7].

Sockeye salmon can be anadromous (ocean-going) or remain as freshwater populations known as kokanee [6]. Populations of sockeye salmon and kokanee can be broadly divided into northwestern and southern phylogenetic groups based on allozyme, minisatellite, and mtDNA loci (a third glacial refugium has also been suggested) [8,9]. This split between northwestern and southern phylogenetic groups are consistent with other Pacific salmon species and suggests two common North American glacial refugia during the Last Glacial Maximum [8–14]). Modern populations of Pacific salmon are thought to be derived from the colonization of fish from these refugia. Kokanee from both phylogenetic groups have diverged from the ocean-going ecotype multiple times (i.e. the kokanee ecotypes are polyphyletic) since the Last Glacial Maximum except in some locations (e.g. the Fraser and Columbia Rivers) [2,8,15], where multiple populations of kokanee are more closely related to each other than sympatric sockeye salmon. At one time the Fraser and Columbia Rivers may have been connected, which could explain how kokanee from these rivers could be monophyletic [15,16].

Several studies have previously identified genomic regions underlying the various sockeye ecotypes (including spawning habitat ecotypes not discussed earlier) [17–23]. As noted in Pritchard et al. (2018), one gene that was identified in some of these studies comparing sockeye salmon and kokanee or other ecotypes (e.g. shore-spawning vs. stream-spawning) was the leucine-rich repeat-containing 9 gene [17,20,21,24]. This gene is proximal to the six homeobox 6 gene that is a candidate gene under strong selection in differing Atlantic salmon (*Salmo salar*) populations (associated with upstream catchment) [24]. Larson et al. (2014 and 2019) also discovered that the MHC class II peptide-binding region in sockeye salmon was under directional selection based on spawning habitat ecotype [18,25].

Sockeye salmon have an XY sex-determination system (with sdY as the sex-determining gene on the Y-chromosome [26–29]). Interestingly, the sockeye salmon Y-chromosome has fused with an autosome making it an X1X2Y system [27,30,31]. The X1 and X2 chromosomes correspond to linkage groups 9b and 9a, respectively (from Limborg et al. (2015) [32]). Not all populations of sockeye salmon appear to have a strong association of sdY to sex [27] suggesting that an alternative sex-determination mechanism(s) may exist in certain populations. SdY-positive females in Atlantic salmon have been identified previously and explained as possible mosaicism, but sdY-negative males are less common and require another explanation [33].

Salmonid sex chromosomes are generally not conserved between genera or species even though linkage groups are often conserved between species otherwise [29,34–36]. In Atlantic salmon, sdY has translocated between chromosomes at least twice since Atlantic salmon speciation and suggests that sex determination in salmon can be "evolutionarily fluid" [37,38]. The sdY gene is surrounded by repetitive sequence and is small, which may allow or possibly facilitate these translocation events and generation of novel sex chromosomes in salmonids [29,34,39].

In this study, we generated the first sockeye salmon reference genome assembly. With the large RNA-seq data sets we produced, the National Center for Biotechnology Information (NCBI) generated a standardized gene annotation of this genome assembly. We also resequenced the genomes of 140 sockeye salmon and kokanee samples from along the northern Pacific Ocean to better understand population structure and genomic loci underlying

divergence between populations and ecotypes. Finally, we were better able to characterize the sex-chromosomes of this species.

## Materials and methods

### Samples

See S1 Methods for sampling strategy.

### DNA extractions, RNA extractions, libraries, and sequencing

DNA was extracted from tissue samples preserved in ethanol (from Eric Taylor's, Scott Pavey's, and Michael Russello's labs) with a DNeasy Blood & Tissue Kit (QIAGEN) following the manufacturer's protocol, or the DNA was already extracted. DNA was extracted from tissue samples (from Fisheries and Oceans Canada and the Freshwater Fisheries Society of British Columbia) preserved in RNAlater (ThermoFisher) following the manufacturer's protocol [40]. RNA was extracted from the Pitt Lake sockeye salmon tissue samples (On170719-1) preserved in RNAlater using a RNeasy Mini Kit (QIAGEN).

Overlapping paired-end library preparation and sequencing (used for genome assembly) was performed at McGill University and Génome Québec Innovation Centre. These libraries were generated following the NEBNext Ultra II DNA Library Prep Kit for Illumina (New England BioLabs). The Pippin Prep (SAGE Science) was used for size selection (peak of distribution: ~488 bp based on 2100 BioAnalyzer (Agilent Technologies)) and the library was sequenced on an Illumina HiSeq 2500 in Rapid mode (PE250).

Mate-pair libraries were prepared at McGill University and Génome Québec Innovation Centre following the Nextera Mate Pair Sample Prep Kit (Illumina). Size selection was performed on the libraries using a 0.5% agarose gel for the following sizes: 3, 5, 10 kbp. These libraries were sequenced on an Illumina HiSeq 2500 (PE125).

For whole-genome resequencing, libraries were produced at McGill University and Génome Québec Innovation Centre using a NxSeq AmpFREE Low DNA Library Kit and NxSeq Adaptors (Lucigen) after DNA passed QC (Quant-iT PicoGreen dsDNA Assay Kit (Life Technologies) and gel electrophoresis). They were then sequenced on an Illumina HiSeq X (PE150) after libraries passed QC (Quant-iT PicoGreen dsDNA Assay Kit, Kapa Illumina GA with Revised Primers-SYBR Fast Universal Kit (Kapa Biosystems), and LabChip Gx (PerkinElmer)).

RNA-seq libraries were generated at McGill University and Génome Québec Innovation Centre after the RNAs passed QC (NanoDrop Spectrophotometer ND-1000 (NanoDrop Technologies, Inc., and 2100 Bioanalyzer (Agilent Technologies)). Total RNA (250 ng) was enriched for mRNA using the NEBNext Poly(A) Magnetic Isolation Module (New England BioLabs), and then cDNA synthesis followed using the NEBNext RNA First-Strand Synthesis and NEBNext Ultra Directional RNA Second Strand Synthesis Modules (New England Bio-Labs). NEBNext Ultra II DNA Library Prep Kit for Illumina (New England Biolabs) was used to finish the library preparation. Libraries were sequenced after passing QC (same as above) on an Illumina HiSeq 4000 (PE 100).

PacBio DNA libraries were prepared with the Pacific Biosciences 20 kb Template Preparation Using BluePippin Size-Selection System instructions at McGill University and Génome Québec Innovation Centre. Briefly, Covaris g-TUBES (Covaris) were used to shear high molecular weight DNA at 4000 RPM for 60 s (each direction). DNA damage repair, end repair, and SMRT bell ligation followed manufacturer's protocol using the SMRTbell Template Prep Kit 1.0 reagents (Pacific Biosciences). Size selection (9 kb-50 kb) was then performed on a

BluePippin system (Sage Science). Sequencing was performed on a Sequel (Sequel Sequencing Plate 2.1, SMRT cells 1M v2).

The 10X Chromium shotgun libraries (for genome assembly) were prepared and sequenced at McGill University and Génome Québec Innovation Centre using the Chromium Genome Reagent kits v2 User Guide RevB protocol after BluePippin size selection for 40 kb–80 kb DNA fragments. The Chromium library was sequenced on an Illumina HiSeq X Ten.

## Genome assembly

The overlapping paired-end 250 bp reads (PE250) and all the reads from the mate-pair libraries were checked for quality using FastQC (default settings) [41]. The PE250 reads were then trimmed using trimmomatic version 0.36 [42] with the following parameters: ILLUMINA-CLIP:TruSeq3-PE-2.fa:2:30:10 (i.e. only adaptors were removed based on review of the output from FastQC). All the mate-pair libraries were also trimmed with trimmomatic (ILLUMINA-CLIP:NexteraPE-PE.fa:2:30:10 LEADING:28 TRAILING:28 SLIDINGWINDOW:4:15 MIN-LEN:50). PacBio long reads were error corrected with the paired-end data using LoRDEC [43] with the following parameters: -k 21 -s 3 -T 50.

Initial contigs and scaffolds were produced using ALLPATHS-LG [44] with the following parameters: Coverage of 64x from the PE250 reads (32x from two lanes), 14x coverage from each of the three mate-pair libraries, genome size set to 2.6 Gbp, ploidy set to one. To increase the contig lengths, a custom pipeline was used to fill gaps in the scaffolds. First, the corrected PacBio reads were aligned to the assembly using BLAST [45] (-task megablast, -evalue 1E-16, -max_hsps 25, -word_size 42, -perc_identity 85, -max_target_seq 4, -outfmt 6). The alignments were then filtered with custom software (S1 File) that filters based on linear alignments (maximum distance between BLAST high-scoring segment pairs (hsps) was 15 kbp, minimum length of all hsps was set to 1000, the minimum average percent identity of all the hsps was 87 with a minimum of 85 for an individual hsps). Each PacBio read was only allowed to have one best location. The script gap_finder.pl from LR_Gapcloser [46] was used to identify the locations of all of the gaps in the assembly. The corrected PacBio reads were then used to fill these gaps if the aligned reads spanned the gap as a single hsps or as two flanking hsps. In either case, the distance of the hsps (one spanning or two flanking) from the gap was only allowed to be 100 bp, the size of the sequence filling the gap was only allowed to be 100x larger than the predicted gap (the prediction was made in the ALLPATHS-LG assembly from mate-pair data), and the minimum gap size to fill was 9 bp.

The assembly was then polished with Pilon [47] using all of the trimmed paired-end data (bwa mem aligned with -M parameter, Samtools [48] sorted, and default Pilon parameters), and then scaffolded using the 10x data with the Arcs/Tigmint/Links pipeline [49–51]. The following parameters were used for the Arcs/Tigmint/Links pipeline: -l 5, -a 0.5, -c 3, -e 30000. After scaffolding with the 10x data, another custom gap filling was performed (same as before except, the maximum distance between hsps was 100 kbp).

The assembly was then error corrected with Arrow [52,53] using the ArrowGrid pipeline [54,55] using all of the PacBio reads before correction (default settings) and then with Pilon again (same as above). Scaffolds smaller than 500 bp were then removed using the seqtk [56] function seq. Finally, the sequences were all made uppercase using Unix commands.

Scaffolds were ordered and oriented onto pseudomolecules/chromosomes using the methodology described in Christensen et al. (2018) [57]. Briefly, scaffolds were aligned to the Atlantic salmon (*Salmo salar*, GCF_000233375.1 [58]), coho salmon (*Oncorhynchus kisutch*, GCF_002021735.1), Chinook salmon (*Oncorhynchus tshawytscha*, GCF_002872995.1 [57]), rainbow trout (*Oncorhynchus mykiss*, GCF_002163495.1 [59]), Arctic charr (*Salvelinus*

*alpinus*, GCF_002910315.2 [60]), and Northern pike (*Esox lucius*, GCF_000721915.3 [61])
genome assemblies using BLAST (-outfmt 6, -word_size 48, perc_identity 94, -max_hsps 100,
-max_target_seqs 10 -evalue 1E-16 for Northern pike: -outfmt 6, -max_hsps 400, -max_tar-
get_seqs 10, -evalue 1E-11). These alignments were then filtered (e.g. off-target or repetitive
elements) using the scripts from Christensen et al. (2018) (-max1/2 0.5, -maxActual1/2 100K,
-minl1 0.25, -minl2 0, -minaln 1K, -avgminper 94, -minper 94 -pidVar 4 for Atlantic salmon
and Arctic charr: -avgminper 93 and -minper 93 for Northern pike: -minl1 0.2, -avgminper 86,
-minper 85). The best placement for each scaffold was then found (-filtMinLen 1K -minWin-
Size 10K -minSizeLarger 10K). Marey maps (graphs with genetic map positions on one axis
and genomic positions on the other axis) were generated for each of the previously published
sockeye salmon genetic maps [32,62,63] using the methodology from Christensen et al. (2018).
The syntenic information was combined with the Marey maps using custom scripts (S1 File).
The scaffolds were then manually ordered in Libreoffice calc using the synteny and Marey
map information. The order of the scaffolds was visualized against other genomes using cus-
tom scripts (S1 File) and ggplot2 [64] in R [65]. A BUSCO [66] analysis was performed for
quality assurance. The actinopterygii_odb9 BUSCO dataset (parameters: -m geno, -c 10 -sp
zebrafish) was used to find the fraction of genes that could be identified in the genome assem-
bly to assess the quality of the assembly.

## Circos plot

Duplicated genomic regions were identified with default settings of SyMap [67] from a modi-
fied copy of the genome fasta file. The lower-case sequences of the genome (sequences masked
by WindowMasker [68] by the NCBI) were first replaced with "N's" using a Unix command
(sed -e '/^>/! s/[[:lower:]]/N/g'). Also, only sequences assigned to chromosomes were retained
with the Samtools faidx command. The orientation of the blocks generated from SyMap was
found using the script Analyze_Symap_Block_Orientation.py from Christensen et al. (2018)
[57]. The percent identity between duplicated genomic regions was found using the output
from SyMap and the Analyze_Symap_Linear_Alignments.py script [57]. The percent of repeti-
tive elements was identified in genomic blocks from the modified genomic fasta file and the
Percent_Repeat_Genome_Fasta.py script [57]. Circos software was then used to generate the
Circos plot [69].

## Gene annotation

RNA-seq libraries (see above) of red muscle (SRA accession: SRX5621463), hind gut
(SRX5621462), stomach (SRX5621461), ovaries (SRX5621460), gill (SRX5621459), spleen
(SRX5621458), pituitary (SRX5621457), white muscle (SRX5621456), pyloric caeca
(SRX5621455), adipose (SRX5621454), heart (SRX5621453), liver (SRX5621452), brain
(SRX5621451), mid gut (SRX5621450), left eye (SRX5621449), upper jaw (SRX5621448), head
kidney (SRX5621447), and lower jaw (SRX5621446) were submitted to GenBank to improve
the NCBI genome annotation. The number of paired-end reads per tissue ranged from 98.7 to
147.5 million.

## Variant calling

The GATK version 3.8 best practices pipeline [70–72] was used as a framework to call
variants on the whole genome resequencing data. First the raw paired-end reads were aligned
to the genome with bwa mem (-M parameter) version 0.7.17 [73], and Samtools sort version
1.9 was used to sort the resulting alignments. The Picard (version 2.18.9) command

AddOrReplaceReadGroups was used to add information about the experiments to the alignment file (validation stringency parameter was set to lenient). Samtools was then used to index the alignment files, and MarkDuplicates from Picard added information about possible PCR duplicates (lenient validation stringency). Samples that were split between sequencing lanes were merged with the MarkDuplicates command and read group information was changed using the Picard command ReplaceSamHeader.

GATK's HaplotypeCaller generated GVCF files (—genotyping_mode DISCOVERY,—emitRefConfidence GVCF) for each individual. These GVCF files were then genotyped using GATK's GenotypeGVCFs command for 10 Mbp intervals (generated by a custom script S1 File). The resulting VCF files were then merged using the GATK command CatVariants. The merged VCF file was then sorted using VCFtools [74] command vcf-sort. The sorted VCF file was then compressed and indexed with the Bgzip and Tabix programs [75].

Variants in the VCF file were filtered using the GATK command VariantFiltration (—filterExpression "QD < 2.0 || FS > 60.0 || SOR < 3.0 || MQRankSum < -12.5 || ReadPosRankSum < -8.0"). This filtered VCF file was then compressed and indexed with Bgzip and Tabix and was used as a training set. SNPs from Veale and Russello (2017), Nichols et al. (2016), and Larson et al. (2016) were used as truth sets (as described in the GATK documentation a truth data set is a set of variants assumed to be real) [17,19,20]. To generate the truth sets, sequences and SNP positions were manually extracted from supplemental files and converted to 1-based positions (instead of 0-based), and mapped to the genome with bwa mem (these SNPs previously did not have genomic positions, but positions relative to a sequenced read). The Samtools sorted sam file was then processed by snp-placer [76] to locate the genomic position of these SNPs (this included a filter step and soft-clipped alignments were removed with command-line tools). The VCF file produced by snp-placer was manually filtered for missing locations and alignment quality scores below 10, and then used to identify the same positions in the unfiltered VCF file produced by GATK using the vcf-isec command from VCFtools (parameters used: -n = 2 -f). The resultant VCF files (for each study compared) contained the intersection of the truth SNPs and the unfiltered variants.

Variant recalibration was performed using the GATK function VariantRecalibrator with the training and truth datasets (parameters used: -resource: training,prior = 12.0, -resource: training,truth,prior = 15.0, -mode SNP, -an QD, -an MQ, -an MQRankSum, -an ReadPosRankSum, -an FS, -an SOR, -an DP -an InbreedingCoeff). After variant recalibration the unfiltered variants were filtered using the ApplyRecalibration function of GATK (—ts_filter_level 99.5 -mode SNP).

Three additional filters were applied depending on the type of analysis used to evaluate the variants/populations. The first filter with VCFtools (parameters:—maf 0.05,—max-alleles 2,—min-alleles 2,—max-missing 0.9,—remove-filtered-all—remove-indels) was used in all analyses. This first filter removed variants that were not biallelic, indels, had missing data in more than 10% of the individuals, that were marked as failed by GATK, or if the minor allele frequency was below 0.05. The second filter additionally removed variants with allelic imbalance (with ratios of the lower count allele to higher count allele less than 0.2, referred to as allele balance filter later) in any of the individuals using custom scripts (S1 File). The final filter additionally removed variants that were in high linkage disequilibrium (LD), which might bias phylogenetic analyses. BCFtools [77] was used to filter variants within a 20 kbp window for high LD, and only allowed two variants within that window to remain with high LD (parameters: +prune, -w 20kb, -l 0.4, -n 2). The number of missing genotypes and average depth was calculated for each individual from the variants after the third filter using a custom script (S1 File).

## Clustering individuals (population stratification)

Three methods were employed to cluster the sockeye and kokanee samples: 1) discriminant analysis of principal components (DAPC) [78,79], 2) Admixture (model-based estimation) [80], and 3) a maximum likelihood analysis (phylogenetic tree) [81]. To reduce the effects of high LD, variants that had been filtered for LD were used in the three clustering methods. DAPC was completed in R with the adegenet [79] and vcfR [82] libraries. Clusters were identified using the find.clusters function (the cluster with the lowest Bayesian information criterion was chosen, 3 in this case), and the optimum number of principal components was found using the optim.a.score function (6 were chosen). Both eigenvalues were retained for the discriminant analysis.

For the admixture analysis, chromosome names were changed in the vcf file using a custom script (S1 File), and the vcf file was converted to an appropriate format using PLINK v1.9 [83,84] (parameters:—double-id,—chr-set 29 no-xy). A cross-validation analysis using the admixture software pointed to a cluster of three having the lowest error, and so a K of three was chosen using default settings. Admixture plots were created in R using the ggplot2 and reshape2 [85] libraries.

The maximum likelihood tree was generated with snpPhylo with a bootstrap value of 1000 and filtering turned off as the data had already been filtered for LD (parameters: -B 1000 -r -m 0.0 -M 0.0). The tree was visualized using the interactive tree of life software [86]. The colour of the groups was chosen to match the DAPC analysis.

## Chromosomal variation underlying population structure

To identify the regions of the genome underlying population structure, eigenGWA (eigen genome wide association) [87] was performed using the egwas command from the GEAR software [88]. EigenGWA identifies which regions of the genome are associated with the given eigenvalues and corrects for genetic drift (via the genomic inflation factor) to identify ancestry informative variants/markers. In this case, the LD1 (similar to PC1 from a PCA) values from the DAPC analysis were used in the eigenGWA. Genomic associations were identified in a pairwise fashion between the three clusters found in the DAPC analysis using variants that were not filtered for LD (second filter). A Bonferroni correction was used to limit false positives ($\alpha = 0.01$ before correction) and peaks were examined only if there were multiple significant variants found in the peak to limit false-positive associations and uncover only the most robust associations (a minimum of five significant variants within 100 kbp of each other). Spurious alignments might cause a single or even a few variants to appear highly significant, but they may not show LD to other proximal variants, which would be expected at short distances.

The admixture ancestry values from Fraser River drainage sockeye salmon (n = 14) and kokanee (n = 12) were also analyzed using eigenGWA. This was done because there was an obvious genetic divergence between the two groups seen from the admixture analysis. The same criteria were used for significance as other eigenGWAs. An eigenGWA was also used to identify loci responsible for a latitudinal cline seen from admixture ancestry values. Again, the same significance criteria were used as above.

## Linkage disequilibrium (LD)

LD ($r^2$) was identified between every variant of a chromosome using VCFtools with the allele balance filtered variants with $r^2$ minimum values of 0.5 (parameters:—geno-r2,—min-r2 0.5). They were visualized in R using the scales [89], ggplot2, and plotly [90] libraries. For regions of interest found from an eigenGWA analysis, the variant with the lowest p-value was used to identify all the markers in that region in LD (minimum $r^2$ of 0.3) using VCFtools. This was

done to simply identify all variants that were in LD with significant GWA and to be able to visualize the genomic distance of this LD block. These variants were extracted from the vcf file using the—positions option of VCFtools. These variants were then visualized with IGV [91].

## Genome-wide association (GWA)

Association tests (logistic regression) were performed using the PLINK v1.9 software with sex and sex-determining gene presence/absence as the traits under investigation. For both phenotypes, the variant set filtered for allele balance was employed, and the DAPC groupings and eigenvalues were used as covariates to account for population structure (parameters—allow-extra-chr,—logistic,—allow-no-sex,—covar). The sex-determining gene, sdY [26], was scored manually (present/absent) from alignments produced for variant calling in IGV for all the individuals. A Bonferroni correction was used to control false positives ($\alpha$ = 0.01).

An association test (logistic regression) was also used to identify regions of the genome associated with ecotype (sockeye salmon vs. kokanee). DAPC values were again used to control for population structure. SNPs with the least stringent filtering were used in this analysis. A permutation test with 1000 permutations was used to identify significance ($\alpha$ = 0.01, with 5 significant variants within 100 kbp of each other).

Finally, an association test was used to identify an alternative sex determination gene(s) for upper Columbia River kokanee that were sdY-negative using the variants with the least stringent filtering. As only 31 individuals were used in this analysis and over 4 million variants were interrogated, it was expected that no variants would pass multiple testing correction. Only the peak with the lowest p-values (with more than five proximal variants with low p-values) is shown for hypothesis generation and for a future candidate gene approach to follow. A permutation test with 1000 permutations was used to assess significance ($\alpha$ = 0.01).

## Individual genomic diversity

Runs of homozygosity were identified from the variants that had been filtered for allele balance using PLINK v1.9 (parameters:—homozyg). The total lengths of the runs of homozygosity were plotted in LibreOffice calc, and tested for significance in R using the aov and TukeyHD functions. The number of heterozygous genotypes and alternative homozygous genotypes per individual were counted from the variants with minimal filtering using a custom script (S1 File). Heterozygotes per kbp was calculated as the number of heterozygous genotypes divided by the total nucleotides in the genome (1,927,125,257) multiplied by 1 kbp. The heterozygosity ratio was calculated as the number of heterozygous genotypes divided by the number of alternative homozygous genotypes [92,93].

## Orthology between species

Orthologous genes were identified between sockeye salmon and two other salmon species, coho and Chinook salmon using the methodology of Christensen et al. (2018) [60]. Briefly, the sockeye salmon genome assembly was individually aligned to the coho and Chinook salmon genome assemblies with BLAST (-task megablast, -evalue 0.000001, -max_target_seqs 3, -max_hsps 20000, -outfmt 6, -word_size 40, -perc_identity 96, -lcase_masking, -softmasking false). The resulting alignment files were filtered with the Compare_Genome_2_Other_Genome_blastfmt6_ver1.0.py (-minl 0.01 -minal 30000) and Filter_Linear_Alignment.v1.0.py (default) Python scripts [60]. NCBI annotated proteins were downloaded for each genome assembly and the sockeye salmon proteins were aligned against the other two protein data sets with BLASTP (-max_target_seqs 3, -max_hsps 20, -evalue 0.01, -outfmt 6). The protein alignment files were then filtered using the Filter_Alignments_Blast_Fmt6_Protein_ver1.0.py script

(-min_per 80, -min_aln_per 80) and orthologs were identified between species using the Orthology_Between_Genomes.v1.1.py Python script [60]. If an orthologous gene was not identified between sockeye salmon and one of the other two species, it was considered missing, but this could occur for several other reasons (e.g. an annotation error in a region of the genome, paralogs obscuring clear orthologous assignment, and poor genome assembly), besides an actual gene loss or gain between species. Missing orthologs were identified with a script (S1 File) and plotted by their genomic positions in R using the ggplot2 library. This was done to identify if there were any regions with an increase in missing orthologs that might indicate a problem with the corresponding region of the sockeye salmon genome assembly.

## Results

### Genome assembly

Before trimming, genome coverage with mate-pair and paired-end Illumina data was ~159x assuming a genome size of 2.4 Gbp. After trimming adaptors and low-quality reads, the coverage dropped to ~87x. PacBio data coverage was ~22x with an average read length of ~7276 bp. The genome assembly was submitted to the NCBI (GenBank assembly accession: GCA_006149115.1, BioProject: PRJNA530256). The metrics reported on the NCBI website were: total sequence length ~1.9 Gbp, number of scaffolds: 38,027, scaffold N50 ~1 Mbp, contig N50 ~330 kbp. The BUSCO analysis identified 88.8% complete, 2.9% fragmented, and 8.3% missing BUSCOs or genes. Like other salmonid genomes [57,58,60], the sockeye salmon genome has extensive homology between duplicated chromosomes (i.e. homeologous regions) generated from the salmonid-specific whole-genome duplication [94] (Fig 1). Some regions retain high nucleotide sequence similarity (> 90%) between homeologous regions after the roughly 90 million years since the genome duplication in an ancestral species [94,95] (Fig 1). Qualitatively, these high sequence similarity homeologous regions appear reduced in length when compared to other salmonids [57,58], but this reduction likely reflects issues with the current assembly quality rather than differences between species (discussed below). It is often difficult to distinguish very high similarity sequences during assembly and these regions tend to remain in small and unplaced contigs.

Regions of high LD were most commonly found around centromeres (Fig 1). Other regions with high LD were found on the sex chromosomes 9a and 9b. There are also large regions with high LD on LG13, LG22, and LG27 which do not appear to be related to centromeres (Fig 1).

### Gene annotation

Over 2 billion reads from 18 tissues were submitted to GenBank for gene annotation of the genome assembly. Standardized NCBI annotation using our submitted RNA-seq data, the reference genome, and other publicly available RNA sequence data was used to identify 38,468 protein-coding genes, 5,185 non-coding genes, and 64,416 fully supported mRNAs (from a total of 9.4 billion reads) [96]. This is less than the 42,483 protein-coding genes identified in Chinook salmon (*O. tshawytscha*) (8 billion reads) [97], 42,884 protein-coding genes in rainbow trout (*O. mykiss*) (7.4 billion reads) [98], and 41,269 protein coding genes in coho salmon version 2 (*O. kisutch*) (7.4 billion reads) [99]. The exact reason for these differences is not known, however, it is likely related to a quality difference between the genome assemblies or differences in RNA-seq data sets. A similar difference between the number of protein-coding gene counts was observed between version 1 and 2 of the coho salmon genome assembly (version 1: 36,425 vs. version 2: 41,269) and likely reflects quality differences rather than any biological reason. This first atlas of annotated genes is still a valuable resource as it provides an important data set for linking genetic variants, phenotypes, and genes.

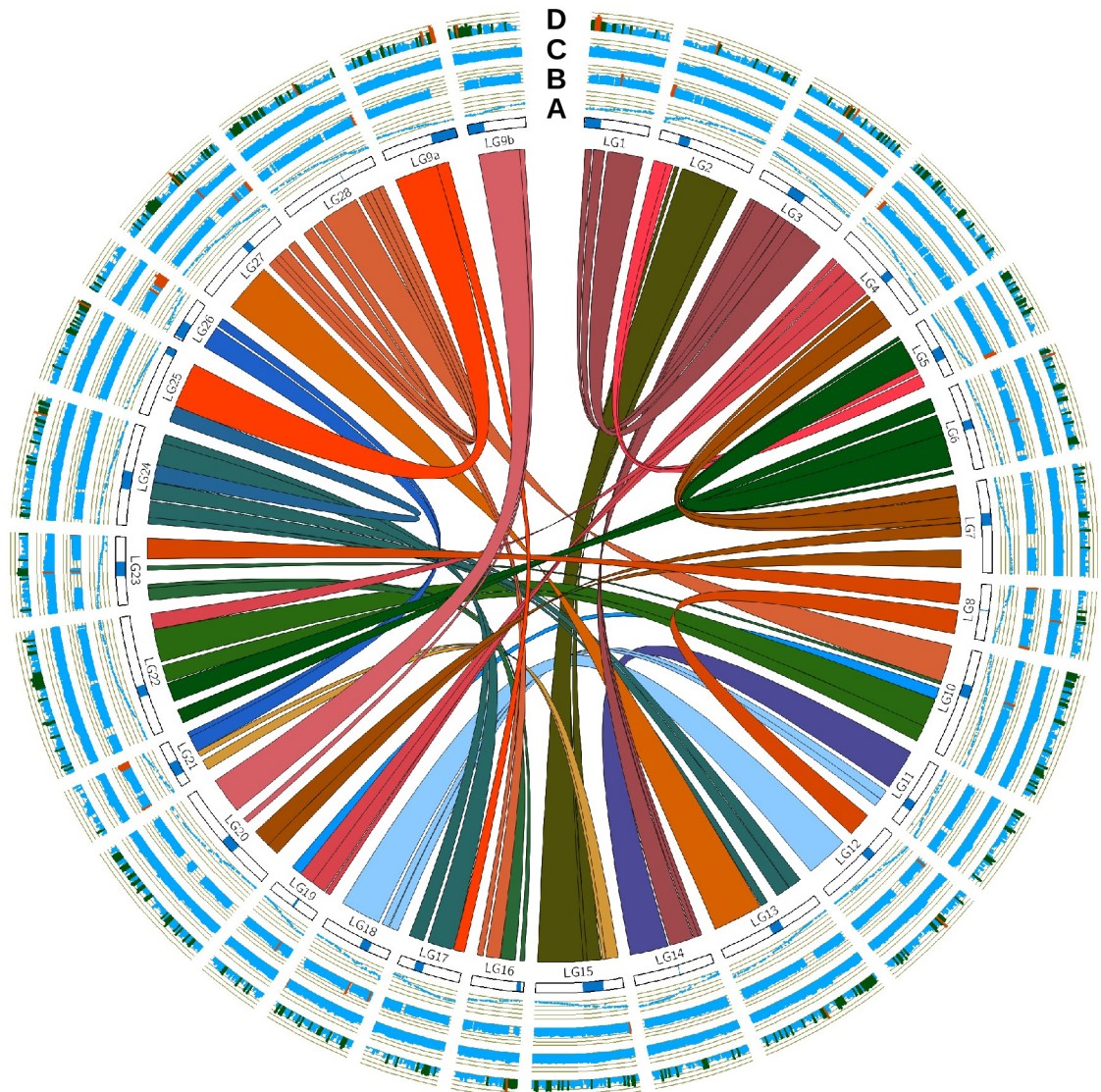

**Fig 1. Sockeye salmon circos plot.** Interior links were generated by SyMap between homeologous regions (only blocks larger than 2 Mbp are shown). Circle A) Larson et al. (2016) [62] female genetic map markers plotted against the corresponding chromosomal positions. Centromeres from Larson et al. (2016) are shown in blue below the genetic map. Circle B) The percent identity between homeologous regions in 1 Mbp intervals and weighted by alignment length (scale: 75–100%). Percent identities above 90% are highlighted with orange. Circle C) The fraction of repetitive sequences in 1 Mbp windows ranging from zero to one (fractions above 0.65 are shown in orange). Circle D) Log-transformed counts of variants with LD ($r^2 \geq 0.5$) to other markers $\geq$ 100 kbp away in 1 Mbp windows. Window counts between 100–999 are green, while those with counts greater than 999 are orange.

## Variant calling

A total of 25,728,393 variants in 140 individuals were filtered to remove indels, variants with more than two alleles, maf < 0.05, and were genotyped in more than 90% of samples to leave 4,533,143 variants. After the second filter (for allele balance), 564,684 variants remained, and after the third filter (LD filter) there were 124,663 variants. The number of uncalled variants and average depth of the variants was calculated from the variants after the third filter. There was an average of 1,866 missing variants per individual with a standard deviation of 2,021. The average depth per variant was 11.10x with a standard deviation of 6.79x.

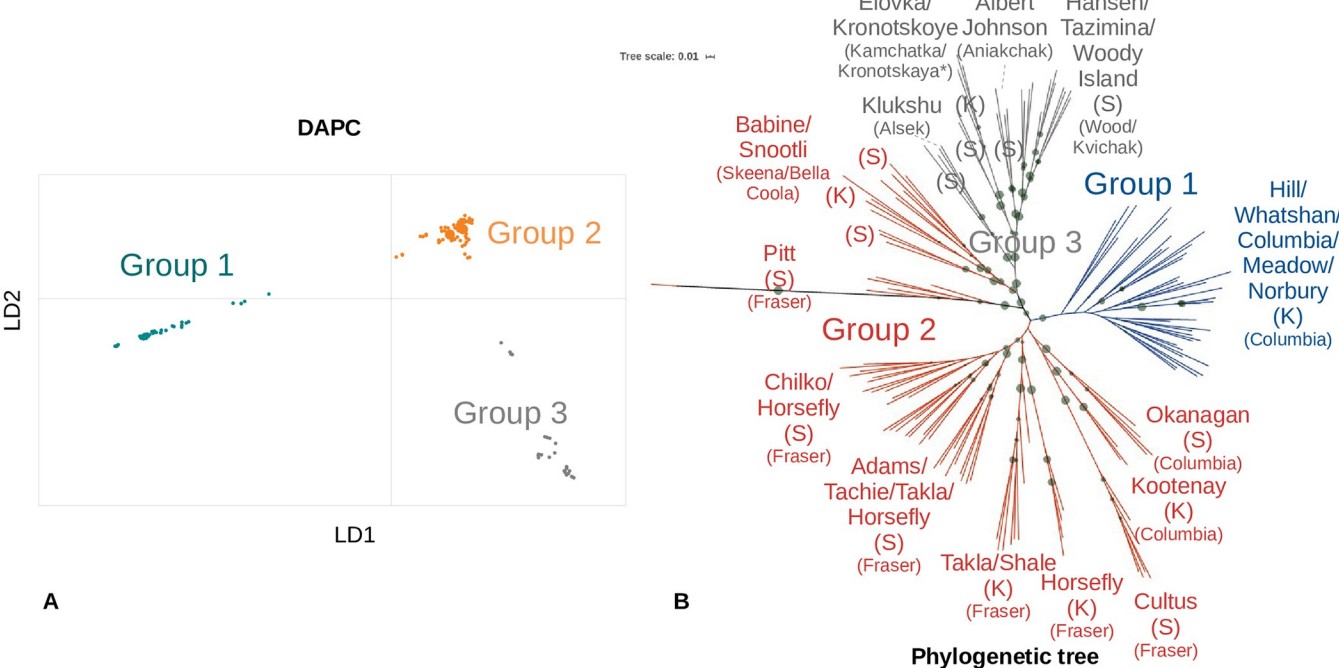

**Fig 2. Clustering sockeye and kokanee individuals (population stratification).** This figure shows the clustering of individuals based on: A) DAPC and B) maximum likelihood analysis (phylogenetic tree). Both analyses began with the same 124,663 variants already filtered for common factors (e.g. maf 0.05) and was specifically filtered for linkage disequilibrium (LD), to reduce the effects of a single genomic location in high LD overwhelming all other signals. A) DAPC analysis clustering with the optimal group number [3] and optimal number of PCAs chosen [6]. The axes represent the first two linear discriminants (LD1 and LD2). The gray, coral (red/orange), and teal (blue) colours correspond to the different clusters. B) An unrooted maximum-likelihood phylogenetic tree, with bootstrap values (based on 1,000 bootstraps) shown as green dots with the larger dots representing greater bootstrap values (min: 0.1 max: 100). Only 7,357 variants remained after default SNPhylo filters. Colours are consistent with the DAPC analysis (DAPC group names shown for comparison). Please note in the Klukshu and Hansen groups, one individual from the other group was found in the grouping and likely represents a switched sample (see S1 Table). Also, there is one Takla kokanee in the Hansen grouping. Sockeye are represented by (S) and kokanee are represented by (K).

## Clustering individuals (population stratification)

DAPC and admixture optimal clustering supported three groups, and the phylogenetic tree appeared to have three main clusters (Figs 2 and 3). DAPC group 1 was comprised of kokanee from the upper Columbia River drainage (Kootenay Lake, Arrow Lake, Whatshan Reservoir, and Koocanusa Reservoir). Please note that the samples from the Clearwater Trout Hatchery, also in DAPC group 1, were from Columbia River ancestry. This group was well supported in every clustering technique. DAPC group 2 was composed of fish from multiple drainages on the BC coast and interior and appears to exhibit clinal variation in admixture values from DAPC groups 1 and 3 (Fig 3). This group also displays variation between sockeye and kokanee in proximal locations from the Fraser River drainage based on admixture ancestry values (Fig 3). DAPC group 3 included all populations either north or west of the Babine River in central British Columbia. The largest uncertainty between the phylogenetic tree and the DAPC analysis was where to differentiate between groups 2 and 3 (Fig 2).

## Chromosomal variation underlying population structure

To identify regions of the genome differentiating groups 2 and 3, an eigenGWA was performed using the LD1 values from the DAPC analysis. The largest and statistically significant signal came from a region of chromosome/linkage group 26 (NC_042560.1 at 26,695,436 bp) that contains a cluster of immunoglobulin heavy chain genes (Fig 4) (referred to as

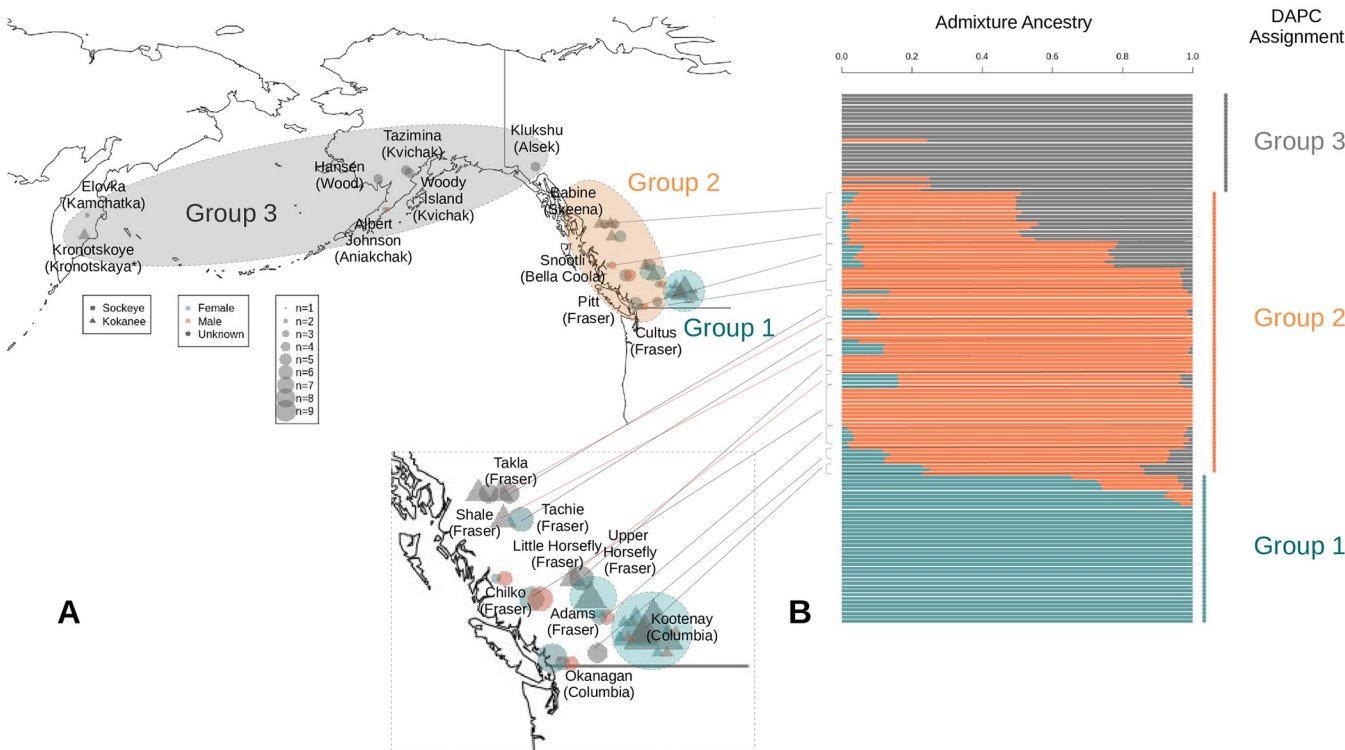

**Fig 3. Population stratification relative to location.** This figure shows the map of the sample sites and the admixture and DAPC clustering analyses. A) The sample site locations with the DAPC assignments overlaid. Specific locations from group 2 are shown with lines. Red lines represent kokanee samples. Only group 2 and 3 body of water names (with drainage in parentheses) are displayed for clarity. The insert shows greater detail of Skeena, Bella Coola, Fraser, and Columbia River bodies of water and is linked by lines to Fig 3B. B) An admixture analysis with k = 3. The colours are consistent with the DAPC analysis and DAPC groups are shown. From the DAPC group 2, there appears to be differentiation between proximal sockeye and kokanee samples based on the admixture ancestry values (in the Fraser River drainage). There also appears to be a latitudinal cline of the gray admixture ancestry values for some sites of DAPC groups 2 and 3.

immunoglobulin heavy chain variable gene cluster). This region shows evidence for large haplotypes, with many individuals being homozygous for a particular haplotype (Fig 4C). However, some individuals are heterozygous for these entire blocks.

The consistent haplotypes and complete heterozygous haplotypes are suggestive of an inversion similar to what has been seen in other salmonid species [100,101]. While there was not an obvious signal of an inversion from aligned reads in this region (e.g. paired-end reads align in the same orientation with a large insert size), there was an excess of paired-end reads with the same orientation in this region relative to the surrounding sequence, possibly indicating gene rearrangements rather than an inversion.

Another statistically significant peak (with at least five variants in peak above α = 0.01) was found on chromosome 16 (NC_042550.1 at 15,084,422 bp) (Fig 4). Calcium channel, voltage-dependent, T type, alpha 1G subunit was identified as a candidate gene for this association.

Eight peaks were significant at α = 0.01 (after Bonferroni correction and with at least five significant variants in the peak) from the eigenGWA analysis between groups 1 and 2 (Table 1, S2 Fig), and ten were found between DAPC groups 1 and 3 (Table 1, S3 Fig). Candidate genes found in these comparisons include: talin 2, "calcium channel, voltage-dependent, T type, alpha 1G subunit", regulator of G-protein signaling 6, dipeptidyl-peptidase 6a, Mtr4 exosome

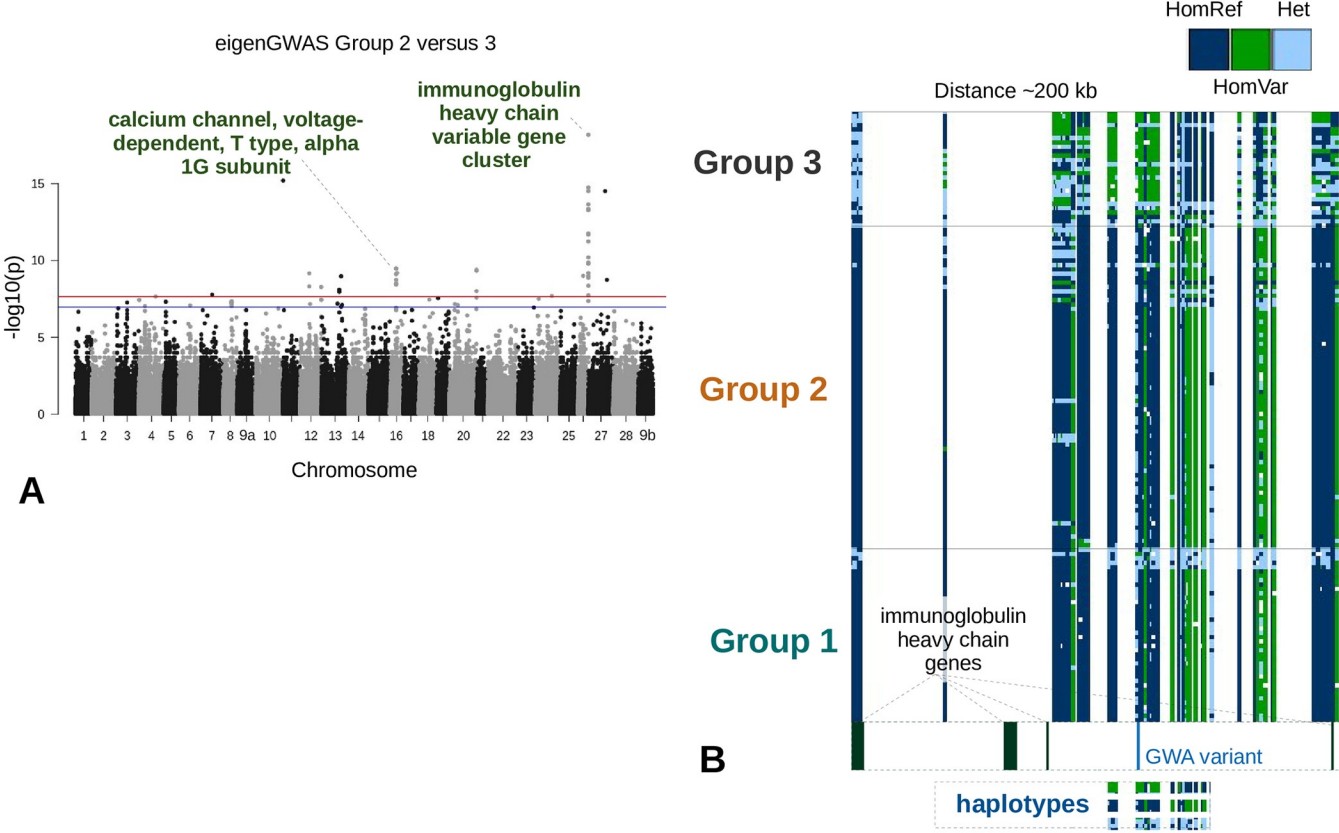

**Fig 4. Genomic regions associated with eigenvalues from DAPC groups 2 and 3.** A) A Manhattan plot of an eigenGWA using the LD1 values from the DAPC analysis after accounting for the genomic inflation factor. Only the individuals from groups 2 and 3 were included in this analysis to specifically find genomic regions underlying clustering differences in DAPC 2 and 3. The horizontal blue line represents the Bonferroni correction at the 0.05 alpha level (variants interrogated = 450,868) and the red line at the 0.01 alpha level (0.01 was chosen as the level of significance for this study). Only peaks with at least 5 significant variants within 100 kbp of each other were considered. B) A screenshot from IGV showing genetic markers in LD ($r^2 >= 0.3$) with the variant with the lowest p-value from the eigenGWA (distance ~200 kb). The dark blue colour represents homozygous reference variants (HomRef), the green colour represents homozygous alternative variants (HomVar), and the light blue colour represents heterozygous variants (Het). Near the bottom of Fig 4B shows the location of the immunoglobulin heavy chain genes and the variant with the lowest p-value are shown. Below that are examples of the haplotypes seen in the data (see Fig 4B legend: dark blue, homozygous for the reference allele; green, homozygous with the alternative allele; light blue, heterozygous). These example data are shown to illustrate that this region is often homozygous for all the alleles in this block.

RNA helicase, "aldehyde dehydrogenase 9 family, member A1a, tandem duplicate 1", GREB1-like protein, and lin-28 homolog B (Table 1).

Some of these associations may be related to inversions found between populations because large haploblocks have been identified in some of these regions. For example, the variant with the lowest p-value in the chromosome 24 peak was found in a haploblock larger than 1 Mbp (S4 Fig). No obvious inversions were seen from aligned reads to this region of the genome. Other mechanisms can generate large haploblocks (e.g. selection, reduced recombination, and inbreeding) and further investigation will be needed to differentiate these mechanisms.

There were nine peaks identified from the eigenGWA comparing the Fraser River sockeye salmon and kokanee (Table 2, S5 Fig). Four candidate genes identified from this analysis were: complement C3-like (LOC115103919), carboxypeptidase A6 (cpa6), cone cGMP-specific 3',5'-cyclic phosphodiesterase subunit alpha'-like (LOC115106380), and SWI/SNF-related matrix-associated actin-dependent regulator of chromatin subfamily A-like protein 1 (LOC115126495). No significant peaks were identified from the latitudinal cline eigenGWA.

**Table 1. Summary of significant eigenGWAS underlying population structure.**

| DAPC 1 vs. 2 | DAPC 1 vs. 3 | DAPC 2 vs. 3 | Chromosome/Scaffold | Position | Candidate Gene Symbol | Accession |
|:---:|:---:|:---:|:---:|:---:|:---:|:---:|
| + | + | | 9a | 20731646 | TLN2 | LOC115134248 |
| + | | | 10 | 61506478 | dcps† | dcps |
| + | + | | 12 | 4475796 | lncRNA† | LOC115138883 |
| | + | + | 16 | 15084422 | CACNA1G | cacna1g |
| | + | | 18 | 31984093 | RGS6 | LOC115146242 |
| | + | | 22 | 5021144 | DPP6A | dpp6a |
| + | + | | 22 | 72378343 | MTREX | LOC115105969 |
| + | + | | 24 | 60038155 | ALDH9A1A.1 | aldh9a1a.1 |
| | + | | 25 | 29879399 | BANP† | LOC115109541 |
| | + | + | 26 | 26674671 | IgHC* | NA |
| + | | | 27 | 20438447 | EMID1† | LOC115111655 |
| + | | | 9b | 29720123 | GREB1L | LOC115114584 |
| + | | | NW_021791234.1 | 120629 | uncharacterized** | LOC115118739 |
| | + | | NW_021786671.1 | 58477 | LIN28B | LOC115116852 |

*Immunoglobulin heavy chain variable gene cluster.

**Similar to immunoglobulin gene.

† Closest candidate gene to lowest p-value variant, but not well supported.

## Genomic associations with kokanee ecotype

Ten loci were associated with the kokanee ecotype (i.e. comparing sockeye salmon and kokanee) (Table 3). With five of these loci it was difficult to identify candidate genes for the following reasons: 1) the association did not overlap with any annotated features, 2) there were no genes on the associated scaffold, 3) there were multiple possible candidate genes, or 4) the genotype information suggested that the association was an artifact of misalignment (personal observation). Five of the associations had a clear candidate gene: neuregulin 3, FKBP prolyl isomerase 6, delta-sarcoglycan-like, and two uncharacterized genes—one that had sequence similarity to an immunoglobulin and the other a non-coding RNA.

**Table 2. Significant eigenGWAS underlying Fraser River kokanee and sockeye ecotypes.**

| Chromosome/Scaffold | Position | Candidate Gene Symbol | Accession |
|:---:|:---:|:---:|:---:|
| 9a | 27555132 | SEPT7† | LOC115134365 |
| 20 | 20896927 | PLXNA2†* | LOC115102444 |
| 21 | 10837054 | C3* | LOC115103919 |
| 22 | 5976979 | CPA6* | cpa6 |
| 22 | 51934852 | P2RX5† | LOC115105525 |
| 23 | 6251055 | PDE6C | LOC115106380 |
| 25 | 30482158 | sspn† | sspn |
| NW_021803831.1 | 67980 | unknown | NA |
| NW_021814461.1 | 17077 | SMARCAL1 | LOC115126495 |

† Closest candidate gene to lowest p-value variant, but not well supported.

* Genes related to ammonia tolerance [102].

**Table 3. Genomic locations of ecotype associations.**

| Chromosome/Scaffold | Position | Candidate Gene Symbol | Accession |
|---|---|---|---|
| 3 | 52034002 | VTCN1† | vtcn1 |
| 4 | 52069691 | PHACTR4† | LOC115128539 |
| 7 | 37872810 | ncRNA† | LOC115132798 |
| 10 | 5469301 | NRG3 | nrg3 |
| 12 | 41953339 | JAG2† | LOC115138579 |
| 22 | 47522975 | FKBP6 | fkbp6 |
| 22 | 50912490 | SGCD | LOC115105509 |
| NW_021813758.1 | 2362 | uncharacterized (ncRNA) | LOC115125940 |
| NW_021814090.1 | 26917 | uncharacterized (diverse immunoglobulin domain) | LOC115126197 |
| NW_021817479.1 | 7319 | unknown | NA |

† Closest candidate gene to lowest p-value variant, but not well supported.

## Sockeye salmon and kokanee sex chromosomes

GWAs were employed to better characterize regions of the genome responsible for sex-determination (Fig 5). Two peaks were observed for phenotypic sex (chromosomes 9a and 9b) and

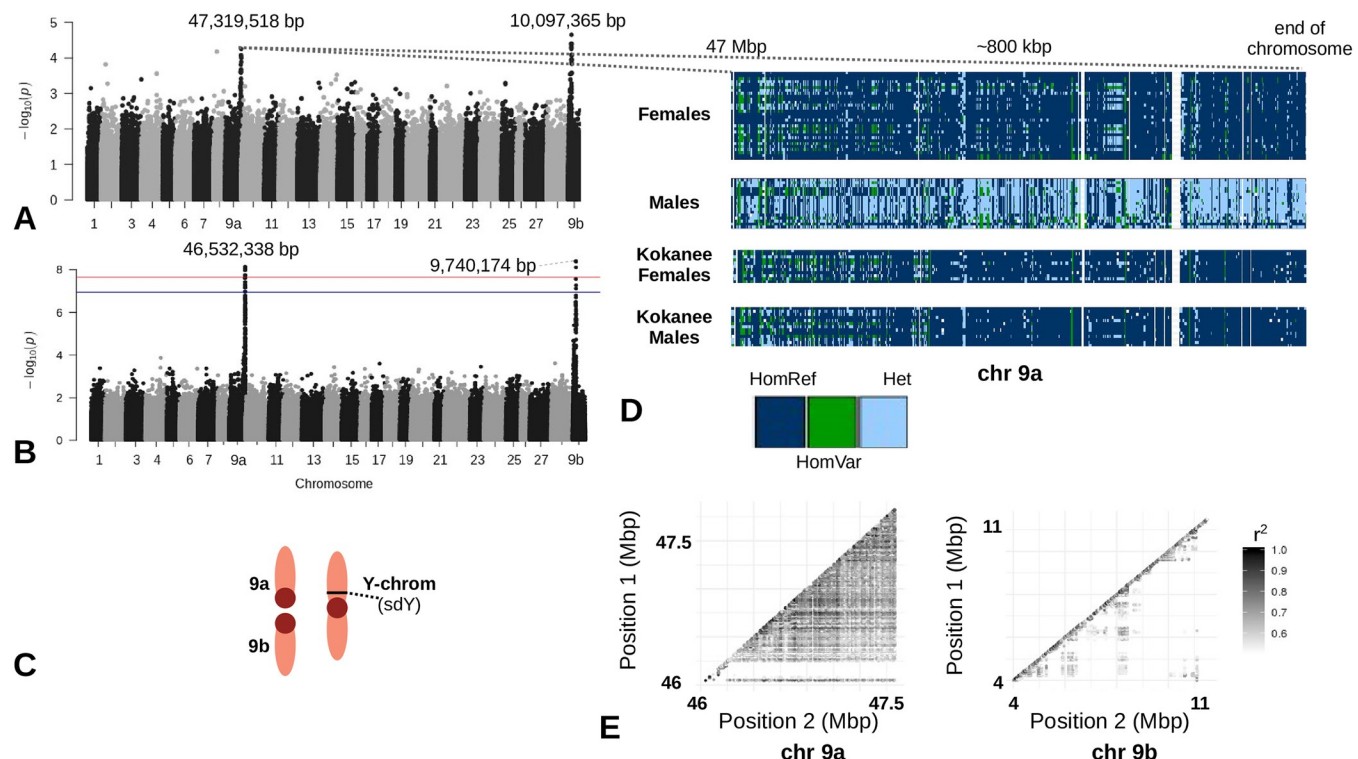

**Fig 5. Sockeye salmon and kokanee sex chromosomes.** A) A GWA between all known male and female sockeye salmon and kokanee salmon with the DAPC LD1 values as covariates. The two peaks (Chr. 9a and 9b) did not reach significance. B) A GWA between all individuals scored for the presence or absence of the sex-determining gene (red line indicates significance threshold). C) A depiction of the proposed X1 (9b), X2 (9a), and Y chromosomes. D) IGV was used to visualize variants on chromosome 9a with only minimal filtering for known females (no upper Columbia River kokanee), males (no upper Columbia River kokanee), Freshwater Fisheries Society of British Columbia female kokanee, and Freshwater Fisheries Society of British Columbia male kokanee. E) On the left, LD plot of chromosome 9a from 4.6 Mbp to the end of the chromosome (~2 Mbp) with a minimum LD threshold of $r^2 = 0.5$ for both plots. On the right, LD plot of chromosome 9b from 3.5 Mbp to 11.5 Mbp.

the sex-determining gene presence/absence GWAs (Fig 5). The associations of the sdY presence/absence analysis but not phenotypic sex reached significance after Bonferroni correction (α = 0.01). These regions on the female sex-chromosomes (as this is a female genome assembly, only the female sex-chromosomes are shown) have extensive LD blocks (Fig 5D).

Another GWA was implemented to identify genomic locations for sex-determination in samples of kokanee from the upper Columbia River drainage (Kootenay Lake, Arrow Lake, Whatshan Reservoir, and Koocanusa Reservoir). These samples were missing the sex-determining gene in male samples. No significant peaks were detected from the GWA. The peak with the lowest p-value from this analysis (multiple variants around krüppel-like factor 5 on LG1 6,248,507 bp) is shown for hypothesis generation and follow-up testing (S6 Fig).

### Individual genomic diversity

The total length of runs of homozygosity ranged from ~6.4 Mbp to ~1.4 Gbp (S7 Fig) with a median of ~35.5 Mbp. The individual with the greatest extent of runs of homozygosity was the artificially produced doubled haploid female. The only population that appeared to have major elevated levels of total lengths of runs of homozygosity was from Cultus Lake (S7 Fig). Total length of runs of homozygosity from Cultus Lake samples were not significantly different with a one-way ANOVA test unless the doubled haploid individual was removed from the analysis (p < 0.001). Samples from all other bodies of water were significantly different from Cultus Lake samples (p < 0.001) using a Tukey's test post-hoc (other comparisons were also significant, but Cultus Lake was the largest difference with all other bodies of water).

The average number of heterozygous genotypes per kbp was 0.67 and varied little between individuals (standard deviation = 0.08, S7 Fig). The heterozygotes/kbp statistic can vary depending on coverage and other factors that were not controlled for. The heterozygous ratio was similar to the heterozygotes/kbp statistic except with the dam of the doubled haploid used to sequence the reference genome (S7 Fig). This individual had much fewer alternative homozygous alleles that inflated the heterozygous ratio. The dam of the individual used in the construction of the reference genome assembly is expected to have lower levels of alternate homozygous alleles as half of its genome was used to generate the reference genome assembly. Excluding this individual, the correlation coefficient between the heterozygous ratio and the heterozygotes/kbp statistic was 0.91.

### Orthology between species

We were able to identify 18,625 orthologs between sockeye and coho salmon and 18,143 orthologs between sockeye and Chinook salmon (S2 File). There were 17,800 coho salmon genes that we were unable to confidently identify an ortholog for and these genes mapped disproportionately to the telomeric ends of coho salmon chromosomes (S8 Fig). The distal segments of salmon chromosomes are often more difficult to assemble due to an ancestral autopolyploidy genome duplication in salmon [94,103]. For this reason, the quality of a genome assembly will likely suffer the most in these regions and may explain why there is a discrepancy of the number of annotated genes between assemblies. Higher-quality assemblies would be able to assembly these regions better and recover gene annotations missed in lower quality assemblies.

## Discussion

### Genome assembly and variant calling

The present sockeye salmon reference genome assembly adds to the growing number of available Pacific salmon genome assemblies. This version of the sockeye salmon genome assembly

has the lowest contiguity metrics and BUSCO scores of the previous Pacific salmon reference genome assemblies. It also has fewer annotated genes and reduced identified orthologs near the ends of chromosomes. Despite these differences, this genome assembly still provides an excellent resource. With this assembly and transcriptome, the NCBI was able to annotate 38,468 protein-coding genes, we were able to identify millions of nucleotide variants, and we were able to identify regions of the genome underlying population structure and the kokanee ecotype.

## Clustering and chromosomal variation underlying population structure

There are three well supported sample clusters, one of which was composed of kokanee samples from the upper Columbia River (Kootenay Lake, Arrow Lake, Whatshan Reservoir, and Koocanusa Reservoir). These results support a closer genetic relationship of kokanee in this region than to sympatric sockeye. This cluster was not found in some previous surveys [8,9,13], but was found in other [15,104]. In one report, the Okanagan population (composed of sockeye salmon and kokanee ecotypes) formed one cluster and all the other bodies of water in the upper Columbia formed a second [104]. This is consistent with results from the current work. In another study, kokanee from central British Columbia (Okanagan Lake, Kootenay Lake, South Thompson River, and middle Fraser River) clustered together while sockeye from this region clustered with sockeye and kokanee from other regions [15]. Taken together, these results suggest the existence of an upper Columbia kokanee population distinct from surrounding populations of kokanee and sockeye (discussed below). Beacham and Withler (2017) suggested that this monophyletic kokanee grouping from central British Columbia was an effect of deglaciation—through the formation of large lakes that connected the Columbia and Fraser Rivers [15,16,105]. We suggest that in addition, there was likely isolation of these kokanee from other sockeye salmon and kokanee that survived in the Columbia River glacial refugia, which is reflected in DAPC grouping, admixture analysis, sex determination mechanism, and a number of genomic regions associated specifically with these kokanee. Isolation of upper Columbia River kokanee (specifically Kootenay River tributaries) has lasted for at least 80 years [106], but isolation might have started potentially up to 10,000 years ago [107,108].

The two remaining clusters are congruent with previous studies, which also found a northwestern and southern grouping [8,9]. Where one grouping ends and the other begins was inconsistent among studies and likely reflects the latitudinal cline seen in the admixture analysis of the current study as well as variation in which specific locations were included in each study. Wood et al. (1994) found a third group in their analysis by splitting the central British Columbia coast from the southern coast [9], which again likely reflects the latitudinal cline seen in the current study.

A major difference between the northwestern group and the southern group was the ancestry informative immunoglobulin heavy chain variable gene cluster found through an eigen-GWA. There are two immunoglobulin heavy chain loci in Atlantic salmon and these duplicated loci likely reflect the salmonid-specific whole-genome duplication [94,109]. In sockeye salmon, the homeologous immunoglobulin heavy chain locus was found on chromosome 21 (~28.3 Mbp—28.6 Mbp) using alignments of the genes on chromosome 26. There were no obvious peaks found on chromosome 21 (Fig 4). As suggested in the Results section, this region might be an inversion between the northwestern and southern groups. Further investigation will be required to better understand this region of the genome and if it is an inversion. Recombination between heterozygous haploblocks is expected to be reduced at an inversion (reviewed in [110]). If the different haplotypes conferred local adaptation (northern vs. southern glacial refugia) that was fixed during glacial isolation, underdominance and lack

of recombination may continue to help maintain population structure between the northwestern and southern groups [110]. Multiple pathogens are influenced by temperature and several outbreaks in fish farms have been related to temperature (reviewed in [111]). It may be that fish from the northwestern group were exposed to a different pathogenic community than those in the southern group due to temperature differences. This region of the genome may reflect that difference in pathogen communities and may confer a selective advantage based on location.

Another significant difference between the northern and southern groups was seen on chromosome 16, near the candidate gene calcium voltage-gated channel subunit alpha1 G. This gene belongs to the T-type calcium channel α1 subunit of the voltage-gated calcium channels that activates in response to low voltage to mediate calcium influx into various excitable cell types [112]. In Atlantic salmon, the T-type and L-type voltage-gated calcium channels were shown to be involved in sperm motility [113,114]. Sperm related genes have been previously found to be under selection between human populations and may represent episodic diversifying selection driven by sperm competition [115].

There were many eigenGWAS peaks between the northwestern and southern groups, and the group from the upper Columbia River drainage. This divergence between the two other groups is likely driven by isolation of this group from the other two for possibly up to 10,000 years. One of the main differences of this group and the two others was an apparent change in sex-determination (discussed below). Other candidate genes identified as ancestry informative between the upper Columbia River group and the other two groups included: talin 2 (TLN2), "calcium channel, voltage-dependent, T type, alpha 1G subunit" (CACNA1G), regulator of G-protein signaling 6 (RGS6), dipeptidyl-peptidase 6a (DPP6A), Mtr4 exosome RNA helicase (MTREX), "aldehyde dehydrogenase 9 family, member A1a, tandem duplicate 1" (ALD-H9A1A.1), GREB1-like protein (GREB1L), and lin-28 homolog B (LIN28B).

Talin 2 is a large gene with multiple transcript isoforms with tissue specific expression, including a spermatid specific isoform [116]. This implicates two possible sperm related genes (talin 2 and calcium channel, voltage-dependent, T type, alpha 1G subunit) in divergence between the three sockeye salmon clusters, possibly associated with diversifying selection driven by sperm competition.

The regulator of G-protein signaling 6 gene is part of a family of regulators that modulate G protein-coupled receptor signaling pathways [117]. It is involved in many biological pathways from cardiovascular development to alcohol dependence in mammals [118,119]. It is unclear why this gene would be ancestry informative in sockeye salmon and kokanee.

The dipeptidyl-peptidase 6 gene helps to regulate cerebellar granuale neuronal cell resting and firing patterns [120,121]. Dipeptidyl-peptidase 6 knockout mice had reduced memory and exhibited slower learning along with brain morphology differences [122]. Interestingly, dipeptidyl-peptidase 6 has previously been found as an outlier locus between a northern and southern snail population in which the author suggested that divergence could be a result of thermally-driven selection [123]. More research would need to be conducted to test this hypothesis in sockeye salmon populations, but we note that sockeye salmon in different geographic regions are subjected to significant variation in thermal habitats.

Mtr4 exosome RNA helicase is a member of the nuclear exome-targeting complex, which functions as a cofactor of the RNA exosome and monitors for aberrant noncoding RNAs [124,125]. The nuclear exome-targeting complex has been found to respond to stress [126], and viruses can co-opt its machinery to initiate viral transcription and increase infectivity [127]. Variants of the Mtr4 exosome RNA helicase might influence infectivity of viruses, and the divergence at this locus may reflect different pathogenic environments and local adaptation of the upper Columbia River kokanee.

The aldehyde dehydrogenase 9 family, member A1a, tandem duplicate 1 candidate gene on chromosome 24 was found in a large haploblock and we suggest that this may be an example of an inversion or multiple inversions between populations. To our knowledge, this is the first putative large inversion found between sockeye salmon populations. Further research will be needed to confirm if this is an inversion or if other mechanisms have driven the relatively large haplotypes in this region of the genome.

GREB1-like protein is a well-known gene that influences the timing of some species of salmonids returning from the ocean to spawn (i.e. run-timing) [128,129]. GREB1-like protein is involved in kidney and genital tract development in mice and zebrafish [130,131], which may help explain the gene's association with run-timing and consequently with maturation in salmon. In sockeye salmon, run-timing and maturation are linked, but maturation can occur in the ocean prior to migration or after migration to freshwater [132]. It may be that the upper Columbia River group is predominately of one run-time and why there is a strong association with this gene. Run-timing data was not collected for this study but should be examined in future investigations now that this association has been identified in sockeye salmon.

Lin-28 homolog B is a key regulator of stem cell self-renewal in many organisms, and the paralog of Lin-28 homolog B plays a further role in primordial germline stem cell development [133]. With its link to spermatogonia, it is possible that, like talin-2 and calcium channel, voltage-dependent, T type, alpha 1G subunit, Lin-28 homolog B could represent diversifying selection driven by sperm competition.

## Genomic associations with kokanee ecotype

In the present study, 19 loci appeared to have an association with ecotype (sockeye vs. kokanee). Four of these associations have previously been identified in other studies [17,19,20]. As discussed in the Introduction, one genomic region that was identified in some association studies comparing sockeye salmon and kokanee was the LG12 region around the leucine-rich repeat-containing 9 gene (start position of gene: NC_042546.1 41,184,975 bp) [17,19,20,24]. This was previously found to be associated with shore vs. stream spawning and sockeye vs. kokanee ecotypes [17,19]. This gene is proximal to the six homeobox 6 gene (start position of gene: NC_042546.1 41,338,065 bp) that is a candidate gene under strong selection in differing Atlantic salmon (*Salmo salar*) populations (24). From the current analysis, the closest ecotype association to this region was at LG12 (NC_042546.1) 41,938,693 bp, which is around 600 kbp away (135 kbp between the lowest p-value variant in the current study and the 22357 RAD tag from Veale and Russello (2017) [19]). Strong linkage disequilibrium, lack of phenotype information (i.e. spawning habitat information that was used in previous research), and much fewer genetic markers in previous studies might be responsible for the distance between peaks. The other three variants previously identified were located on LG20 (kokanee vs. sockeye, RAD tag = 24539 [17] distance from lowest p-value = 153 kbp), LG21 (kokanee vs. sockeye, RAD tag = 91349 [19], distance from lowest p-value = 93 kbp), and LG25 (kokanee vs. sockeye, RAD tag = 58166 [17], distance from lowest p-value = 6 kbp). These correspond to two candidate genes we are not confident about and complement C3-like (on LG21, discussed below). The association identified on LG20 was closest to a possible candidate gene, plexin-A2-like (LOC115102444), discussed below. The variant with the lowest p-value on LG25 was within the sarcospan gene (sspn, discussed below).

One of the conserved associations (found with the GWA between all sockeye and kokanee) in the current study was with the candidate gene neuregulin 3. Neuregulin 3 is a member of the epidermal growth factor-like signaling molecule family of genes and plays a role in the central nervous system development (reviewed in [134]). It has been associated with various

behaviours and psychiatric disorders in humans and mice [134]. It is unclear at this time how this gene might influence ecotype or why it is associated with ecotype.

Delta-sarcoglycan, another conserved candidate gene (i.e. an association identified in the analysis between all sockeye and kokanee), is a component of the sarcoglycan subcomplex that stabilizes skeletal muscle fiber sheaths among other functions (reviewed in [135]). Variants of this gene are associated with muscular dystrophy in humans [136], and this gene plays a role in the longevity of the retina in mice [137]. The sacrospan gene, another candidate gene, associates with the sarcoglycan subcomplex [138]. Proteins from these genes are components of the sarcoglycan-sarcospan complex and are expressed in the retina, likely in Müller and ganglion cells [139]. How the sarcoglycan-sarcospan complex might relate to sockeye salmon ecotype requires a brief explanation of smoltification.

Smoltification is critical to ocean-going sockeye salmon as it prepares the parr (developmental stage before smoltification) for the challenges of a marine environment. Unlike most sockeye salmon, kokanee remain in fresh-water environments. Smoltification alters metabolism, osmoregulation, growth, colour, behaviour, and other traits of the young parr to prepare for marine environments (reviewed in [140]). In landlocked Atlantic salmon (analogous to kokanee), some populations still have many of the elements of smoltification while others have lost key components, including osmoregulatory ability, brain structure development, and metabolism [140]. Kokanee can go through smoltification, but like landlocked Atlantic salmon, it appears that the process has been altered from ocean-going sockeye salmon in at least one population [107]. Smoltification is energetically costly and is possibly maladaptive in landlocked salmon [140].

One of the environmental cues for smoltification comes from changes in day length detected by the retina and the light-brain-pituitary axis—changes to this system such as continuous daylight may interrupt smoltification [140]. Activation of the light-brain-pituitary in some landlocked Atlantic salmon smolts by day length appears to be disrupted [140]. One possible explanation for this disruption is that it reduces the chance of smoltification in these landlocked populations and offers an evolutionary advantage because energetic resources are not used in a process no longer needed. If the sarcoglycan-sarcospan complex is involved in maintaining the retina in sockeye salmon, then it may play an indirect role in smoltification and in a similar disruption of the light-brain-pituitary axis.

Another gene associated with sockeye ecotype for a subset of the samples was the cone cGMP-specific 3',5'-cyclic phosphodiesterase subunit alpha' gene (PDE6C). This gene is a vital component of the phototransduction pathway (reviewed in [141]) and is differentially expressed in the brains of resident and migratory rainbow trout along with many other phototransduction genes [142]. It has also been previously identified as under local selection in Atlantic salmon [24] or associated with domestication (as indicated in Pritchard et al. 2018 [24,143]). Again, this gene may play a role in smoltification and the disruption of the light-brain-pituitary axis, which may be favourable for kokanee and landlocked populations.

Three candidate genes: complement C3, carboxypeptidase A6, and plexin A2 were previously identified as candidate genes for ammonia tolerance in orange-spotted grouper (*Epinephelus coioides*) [102]. All three of these candidate genes were established in previous studies identifying outlier loci between kokanee and sockeye populations (plexin A2-like and complement C3 outlier loci were within 200 kbp of the current associations and the carboxypeptidase A6 outlier locus was within 1 Mbp, RAD tag: 14532 [17]). These candidate genes and their link with ammonia tolerance, suggests that a driving force in divergence between Fraser River sockeye salmon and kokanee might be environmental ammonia. Sockeye salmon experience varied levels of environmental ammonia, which is expected to be highest in estuaries where run-off from agriculture and other human activities accumulates [144]. Sockeye salmon

smolts, out of all the Pacific salmon, spend the least amount of time in estuaries—only around 5 days [145]. If estuary ammonia levels are responsible for the divergence between sockeye salmon and kokanee at these loci, it suggests that these are recent and human-induced adaptations. Further research will be needed to test this hypothesis.

Similar to differences between the DAPC groups, a sperm-related gene differentiates sockeye and kokanee ecotypes. The FKBP6 prolyl isomerase protein functions as a testis-specific component of the synaptonemal complex and is essential for sperm production in mice [146]. This region has previously been identified as an outlier locus between Okanagan Lake kokanee and Okanagan River sockeye salmon and also between Redfish Lake kokanee and sockeye (within 1 Mbp—RAD tag: 54636 [19], RAD tag: 47864, [17]). Again, this result suggests diversifying selection driven by sperm competition.

## Sockeye and kokanee sex chromosomes

The Y-chromosome in sockeye salmon is a metacentric chromosome formed from the fusion of the acrocentric Y-chromosome with another acrocentric autosome [31]. The sex phenotype maps to the putative centromere of this fusion [27]. In females, the X1 and X2 chromosomes correspond to chromosomes 9b and 9a respectively [32]. Both female chromosomes show association with the sex phenotype, sdY (sex-determining gene in salmon [26] and sockeye salmon [27]), and have large haploblocks based on sex in this study. This is consistent with a Robertsonian translocation and reduced recombination common to sex-determining regions.

Male kokanee from the upper Columbia River drainage were sdY-negative (personal communication DS and RHD), which is consistent with the resequencing data aligned to the sdY sequence. It is also congruous with a previous study finding a high percentage of sdY-negative sockeye salmon males (~30%) in an upper Columbia River hatchery [27], and another study with similar findings in samples collected from Asian populations [147]. Atlantic salmon females have been identified with sdY, but likely have autosomal pseudocopies rather than a bonafide function sex-determining copy (bioRχiv [148]) [33]. This has been noted in other salmonid species as well [147]. There were no significant associations from a GWA of the sex phenotype with any markers, but there was a ~100 kbp peak at the krüppel-like factor 5 gene (LG1 6,248,507–6,256,452 bp). This peak encompassed the markers with the lowest p-values with the sex phenotype. Future studies of mid or upper Columbia River sockeye salmon and kokanee will be needed to better our understanding of alternative sex determination in salmonids. From this and a previous study, there is a clear alternative sex-determination mechanism to the canonical sdY pathway in a potentially large percentage of upper Columbia River sockeye salmon and kokanee.

The sdY gene arose from a gene duplication of an immune-related gene that diverged to be able to interact with the Forkhead box domain of the female-determining transcription factor and eventually disrupt female differentiation [149]. Thus, sdY was able to "hijack" a conserved sex differentiation cascade by interacting with one of the members of this cascade [149]. Krüppel-like factors (Krüppel-like factor 4 directly) have been shown to interact with this differentiation cascade [150,151] and they could have likewise been co-opted to serve as sex-determining genes in kokanee.

## Individual genomic diversity

Overall, genomic diversity was similar among all samples except in a doubled haploid individual, the dam of the individual used in the construction of the reference genome assembly, and samples from Cultus Lake. The Cultus Lake population is considered endangered and has declined in abundance precipitously since the 1970s [152]. Previously, Cultus Lake samples

had the lowest mean heterozygosity scores (0.57) compared to other Fraser River drainage samples, which otherwise had uniform heterozygosity scores in one study examining six microsatellite markers [153]. This is consistent with the high total length of runs of homozygosity found from the Cultus Lake samples in this study that were not seen in any other bodies of water. These baseline metrics of genomic diversity may play an important role in discussions of conservation of threatened populations of sockeye salmon.

## Conclusions

In this study, we generated a reference genome assembly for sockeye salmon, a useful RNA-seq data set for annotation of this and future sockeye salmon genome assemblies, and identified regions of the genome underlying population structure and sockeye salmon ecotype. We found that an immunoglobulin heavy chain locus was a major ancestry important region of the genome differentiating two of the three key genetic groups of sockeye salmon and kokanee from this study. We were able to identify regions of the genome that appear to differentiate sockeye salmon from kokanee, and these regions implicate ammonia tolerance and vision as possible indicators of ecotype. Finally, we were able to improve understanding of the sex chromosomes in sockeye salmon, and to confirm a novel sex determination mechanism in kokanee.

## Supporting information

**S1 Fig. Sample site locations of sockeye salmon and kokanee.** Map generated with the maps library in R [154].
(TIF)

**S2 Fig. EigenGWA between the DAPC groups 1 and 2.** A Manhattan plot where eigenvalues from the DAPC analysis were used to identify regions of the genome with ancestry informative genes (e.g. under selection) between groups 1 and 2. The red horizontal line is the threshold of significance for α = 0.01 after Bonferroni correction. The blue line is for α = 0.05.
(TIF)

**S3 Fig. EigenGWA between the DAPC groups 1 and 3.** A Manhattan plot where eigenvalues from the DAPC analysis were used to identify regions of the genome with ancestry informative genes (e.g. under selection) between groups 1 and 3. The red horizontal line is the threshold of significance for α = 0.01 after Bonferroni correction. The blue line is for α = 0.05.
(TIF)

**S4 Fig. Chromosome 24 (NC_042558.1) genotypes and putative inversion(s).** A) On the left of this figure is the admixture ancestry plot with the DAPC group assignments. On the right, is a screenshot of chromosome 24 from IGV from 58 Mbp—62 Mbp (only variants with $r^2$ values $>= 0.3$ with the variant with the lowest p-value from the eigenGWA in this peak are shown). This region of the genome was found from an eigenGWA to be associated with inferred population structure between DAPC groups 1 and 2. The dark blue genotypes are homozygous for the reference allele (HomRef), the green genotypes are homozygous for an alternative allele (HomVar), and the light blue are heterozygous (Het). B) A scatterplot of variants with $r^2$ values $>= 0.5$ on the top shows areas with high LD. Below is a smaller version of the genotypes with the putative inversions highlighted.
(TIF)

**S5 Fig. Sockeye salmon vs. kokanee eigenGWA.** A) The eigenGWA is shown between Fraser River sockeye salmon (n = 14) and kokanee (n = 12) with putative genes highlighted at the

peaks (with at least 5 variants with LD). The red line represents a Bonferroni correction at α = 0.01 and after correction for the genomic inflation factor. The blue line represents a Bonferroni correction at α = 0.05 and was chosen as the minimum value of significance. B) An IGV plot of all the variants used in the eigenGWA for the region around the peak on chromosome 23. The genotypes are: dark blue—homozygous reference, green—homozygous alternative, and light blue—heterozygous. The top IGV plot is the kokanee used in this analysis and the sockeye are below. Below the IGV plot, thick lines represent NCBI annotated genes in this region. The putative ancestry informative gene is highlighted in green and named. The variants with the lowest p-values from the eigenGWA are shown as dotted-lines (1st represents the variant with the lowest p-value, 2nd represents the variant with the second lowest p-value, etc.). The p-values, in combination with the genotypes, were used to identify the most likely ancestry informative gene in this region.
(TIF)

**S6 Fig. Visualization of the variants with the greatest association to the sex phenotype in kokanee lacking the sdY gene.** Variants on chromosome 1 (NC_042535.1) shown in IGV with the female variants on top and the male variants on the bottom. The variant with the greatest association was found in the 3' UTR of the krüppel-like factor 5 gene.
(TIF)

**S7 Fig. Individual genomic diversity.** A) A map of the sampling sites. B) Three measures of individual genomic diversity: 1) total length of runs of homozygosity, 2) heterozygous genotypes per kbp, and 3) heterozygous ratio.
(TIF)

**S8 Fig. Density plot of genes that we were unable to identify an ortholog for on the coho salmon genome.** The x-axis is positions along the chromosome and the points represent the start position of a "missing" gene. The y-axis is the density of missing genes along the chromosome.
(PDF)

**S1 Table. Sample information.**
(XLSX)

**S1 File. Compressed archive file with various custom Python scripts used in this study and readme files.**
(XZ)

**S2 File. List of orthologous genes between sockeye salmon and other salmon species.**
(XLSX)

**S1 Methods. Sampling strategy.**
(DOCX)

## Acknowledgments

We would like to acknowledge and thank McGill University and Génome Québec Innovation Centre for their extensive sample preparation and sequencing services. We would also like to thank and acknowledge the generous support and resources from Compute Canada (www. computecanada.ca). We would also like to thank Fisheries and Oceans Canada and the University of Victoria for the facilities and personnel needed for this study. Finally, the authors also appreciate the many Fisheries and Oceans Canada research and watershed management staff who collected samples for analysis in this study.

## Author Contributions

**Conceptualization:** Kris A. Christensen, Eric B. Rondeau, David R. Minkley, Robert H. Devlin, Ben F. Koop.

**Data curation:** Kris A. Christensen, Dionne Sakhrani, Anne-Marie Flores, Theresa Godin, Eric B. Taylor.

**Formal analysis:** Kris A. Christensen, Eric B. Rondeau.

**Funding acquisition:** Robert H. Devlin.

**Investigation:** Kris A. Christensen, Theresa Godin, Robert H. Devlin, Ben F. Koop.

**Methodology:** Kris A. Christensen, Eric B. Rondeau, David R. Minkley, Dionne Sakhrani, Carlo A. Biagi, Anne-Marie Flores, Ruth E. Withler, Ben F. Koop.

**Project administration:** Kris A. Christensen, Dionne Sakhrani, Robert H. Devlin, Ben F. Koop.

**Resources:** Dionne Sakhrani, Carlo A. Biagi, Ruth E. Withler, Scott A. Pavey, Terry D. Beacham, Theresa Godin, Eric B. Taylor, Michael A. Russello, Robert H. Devlin, Ben F. Koop.

**Software:** Kris A. Christensen, David R. Minkley.

**Supervision:** Kris A. Christensen, Robert H. Devlin, Ben F. Koop.

**Validation:** Kris A. Christensen, Anne-Marie Flores, Ruth E. Withler.

**Visualization:** Kris A. Christensen.

**Writing – original draft:** Kris A. Christensen.

**Writing – review & editing:** Kris A. Christensen, Eric B. Rondeau, Anne-Marie Flores, Terry D. Beacham, Theresa Godin, Eric B. Taylor, Michael A. Russello, Robert H. Devlin, Ben F. Koop.

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
