## [Decision Letter · Decision Letter 0]

16 Jun 2020

PONE-D-20-12784

The sockeye salmon genome, transcriptome, and analyses identifying population defining regions of the genome and sex chromosome characterization

PLOS ONE

Dear Dr. Christensen,

Thank you for submitting your manuscript to PLOS ONE. After careful consideration, we feel that it has merit but does not fully meet PLOS ONE’s publication criteria as it currently stands. Therefore, we invite you to submit a revised version of the manuscript that addresses the points raised during the review process.

Your manuscript has been reviewed by three referees. Although the external referees express interest in the general subject area of the paper, they also express a series of reservations that preclude publication of the paper in PLoS ONE in its current form. However, if you feel that you can suitably address the concerns and issues raised by the referees, I would be willing to consider a revised manuscript. Also, please be advised that the revised manuscript may be subject to re-review.

We look forward to receiving your revised manuscript.

Kind regards,

Zuogang Peng, Ph.D.

Academic Editor

PLOS ONE

Journal Requirements:

2. To comply with PLOS ONE submissions requirements, please provide methods of sacrifice in the Methods section of your manuscript.

3. We note that you are reporting an analysis of a microarray, next-generation sequencing, or deep sequencing data set. PLOS requires that authors comply with field-specific standards for preparation, recording, and deposition of data in repositories appropriate to their field. Please upload these data to a stable, public repository (such as ArrayExpress, Gene Expression Omnibus (GEO), DNA Data Bank of Japan (DDBJ), NCBI GenBank, NCBI Sequence Read Archive, or EMBL Nucleotide Sequence Database (ENA)). In your revised cover letter, please provide the relevant accession numbers that may be used to access these data. For a full list of recommended repositories, see http://journals.plos.org/plosone/s/data-availability#loc-omics or http://journals.plos.org/plosone/s/data-availability#loc-sequencing.

4. In your Methods section, please provide additional location information of the sampling sites, including geographic coordinates for the data set if available.

Reviewers' comments:

Reviewer's Responses to Questions

**Comments to the Author**

1. Is the manuscript technically sound, and do the data support the conclusions?

Reviewer #1: Yes

Reviewer #2: Yes

Reviewer #3: Partly

2. Has the statistical analysis been performed appropriately and rigorously? 

Reviewer #1: Yes

Reviewer #2: Yes

Reviewer #3: Yes

3. Have the authors made all data underlying the findings in their manuscript fully available?

Reviewer #1: Yes

Reviewer #2: Yes

Reviewer #3: Yes

4. Is the manuscript presented in an intelligible fashion and written in standard English?

Reviewer #1: Yes

Reviewer #2: Yes

Reviewer #3: No

5. Review Comments to the Author

Reviewer #1: In this MS, the authors sequenced and assembled the first sockeye salmon reference genome assembly. The genomes of 140 sockeye salmon and kokanee from various bodies of water along the northern Pacific Ocean were resequenced to understand population structure and genomic loci underlying that population structure. Three distinct groups were identified from the individuals. An immunoglobulin heavy chain variable gene cluster on chr. 26 was identified that differentiated the samples from the northwestern region of the sampling area from those to the south. They also explore the sex chromosomes of this species. An alternative sex-determination mechanism was identified in a subset of upper Columbia River kokanee.

Generally, the manuscript is well organized and written nicely. The methods were clearly described. The data analysis and discussion were appropriately made. The figures are appropriate and clear. I have just a few issues that need to be addressed.

1.Line 374-376, please add references.

2.Line 372, the sockeye salmon, sockeye salmon,and coho salmon are closely related species. Why the protein-coding genes annotated in this species were much less than salmon and trout? Whether this indicate the poor quality of the genome assembled or some characteristics related to this species? Please discuss it in detail. In addition, what genes were not identified in this species?

3.The authors identify the Chr.9a/9b as the sex chromosomes. However, no data was provided to support this conclusion because no marker was developed based on the differentiated regions on this chromosome to distinct the genetic male and female. The sockeye salmon has XX/XY or ZZ/ZW sex determination system? Please indicated it in the introduction.

4.The authors analyze the genomic associations with kokanee ecotype. Some candidate genes, including aquaporin-3, trim45, etc, were identified and discussed in three populations. But no functional experiments were performed to demonstrate it. In addition, whether the expression of these genes showed different patterns in different populations?

5.In Figure 2B, this figure was used in Figure 3, 4, 5 and 6. Please delete it.

6.Species name in the References should be in italics.

Reviewer #2: Christensen et al. describe a reference genome for sockeye salmon and conduct genomic analysis of sockeye salmon from across their range. The resource is highly valuable, the population genetic results are compelling, and the paper was well written. I recommend that the paper be accepted with very minor revisions. I only have a few comments.

I thought the finding of the large region of divergence between lineages on chromosome 26 was very interesting. I’m wondering if this could be an inversion or if some other mechanism is contributing to the large peak. Maybe this could be similar to inversions between lineages in Atlantic salmon? (e.g. https://doi.org/10.1111/mec.15065) I recommend the authors add a few sentences discussing potential mechanisms that could explain this large peak.

Line 330: I think “covariants” should be “covariates”

Line 419: Hanzen should be Hansen

Line 688: loci to locus

Line 716: I thought the inclusion of the double haploid in the runs of homozygosity analysis was unnecessary. I recommended removing this individual from the analyses and taking the section describing these results out of the discussion.

Reviewer #3: Comments to authors

This manuscript describes the acquisition and analyses of genome sequence data from a large number of sockeye salmon from throughout the species range. In addition, RNA-seq data were generated to help with annotated the composite genome. The genome data were then analyzed for polymorphisms that were used to help answer questions important to the evolutionary ecology of sockeye salmon. The authors have generated a lot of sequence data and conducted many analyses. Constructing a genome sequence, especially in a species with residual tetrasomy, is challenging. however, I have many concerns both with regard to the writing and clarity of the manuscript as well as the analyses chosen. Below I will split up my concerns and comments into what I deem to be major revisions and those which are more minor in nature.

Major comments

1) I feel that the manuscript suffers from an identity crisis. Is the main message to present the genome/protein coding parts of the genome or is the main aim of the manuscript to describe the analyses of the genetic data to answer the genetic basis of interesting questions? These needn’t be mutually exclusive, but I think the manuscript would read better if the authors could focus the writing on one of these two big picture aims.

2) The writing is poor and in places very difficult to read. Please give the manuscript a thorough re-read before resubmitting. Below in the minor comments I will detail some specific line numbers and sentences that were especially difficult for me to follow.

3) The quality of the figures MUST be addressed. The DPI was too low to accurately read figures and this greatly diminished my enthusiasm for the manuscript. In addition, many of the figures seemed unnecessary and I would recommend moving to the supplemental data.

4) Some analyses should be completed before resubmission. Specifically, I would like to see more information on how the authors dealt with the homeologous regions of the sockeye genome. Where they removed from this assembly? A circos plot, similar to previous salmonid genomes, would be very helpful in that regard.

Minor comments/specific concerns

Abstract

Lines 28-30, arguably, this is a common feature of many different salmon and trout (e.g., rainbow trout and chinook). Maybe remove the “most complex and fascinating life histories” and instead say that “Repeatedly, a resident form known as kokanee…” and continue with the rest of lines 29-30.

Line 34: do a better job of linking these sentences, the polymorphisms within the immunoglobulin heavy chain are what’s causing a large part of the differentiation between the three groups.

Introduction

Paragraph starting on line 52 is confusingly written and much of the information seems unnecessary. The take home message is that sockeye exist as multiple different life history ecotypes and one of these is the freshwater resident kokanee. I’d recommend deleting this paragraph, taking the important message (i.e., what I say above) and adding this to the paragraph starting on line 61.

Line 65: I’d suggest replacing “the hypothesis of…” with “is believe to be due to two common North American…”

Lines 69-71: this sentence must be clearer! The way I read it the kokanee appear to be monophyletic with respect to multiple rivers from the same area. Is that correct?

Lines 75-76: selection with respect to what in Atlantic salmon? As in how is variation in this gene associated with selection?

Line 81: delete “assembly”.

Lines 83-84: delete “various bodies of water”.

Lines 84-85: replace “that population structure” with ecotype divergence.

Materials and methods

Samples

For the two fish used in genome sequencing and transcriptome sequencing please state what population these samples originated from. How old were the samples?

Line 108/ figure 1: i’d recommend removing figure 1 as it is difficult to determine how many samples from each population there are and instead displaying that information as a table with the location, latitude and longitude, sample size, number of each sex, and the number of kokanee versus anadromous sockeye. This would be a useful resource for the reader as they progress reading the manuscript.

Lines 137-142: I am guessing that the samples were barcoded to allow pooling when sequenced? I’m assuming yes, but I can’t find the specific details in the methods either here or in the variant calling section. Some mention for how samples were barcoded and how sequences were separated by sample should be made.

Line 144: delete “that were sent”.

Line 147: the NEBNext RNA first strand synthesis is a way of synthesizing cDNA not for enriching extracted RNA for mRNA.

Line 165: what quality filters were used?

Line 169-170: I don’t understand the “and using paired-end data” add on. I have a feeling this should be a separate sentence.

Line 174: “corrected” PacBio reads, should that be filtered or quality filtered?

Line 198: “found” in Christensen et al should be “described”.

Line 205: why were alignments filtered? What were the authors trying to remove?

Line 210: please add sockeye to “previously published genetic maps”.

Line 248: please replace “truth datasets” with to validate candidate SNPs.

Line 249: please replace “the truth without errors” with real.

Line 281-283: this sentence needs a rewrite. I’d recommend “Each of the methods used filtered variants to reduce the effects of high LD on subsequent analyses”.

Line 299-300: “from the clustering methodologies” (see clustering individuals section) should be deleted.

Lines 304-307: I’m confused by what was done here and why. What do the authors mean by “allele balance”? why wasn’t LD filtered?

Line 307: what was the p-value cutoff for the Bonferroni correction?

Lines 310-312: I am confused as to what data were used for this analysis and why.

LD section

Please check the superscript formatting of R2 it seems off.

I’m concerned about filtering for minimum R2 value around regions that might be in high LD. Would doing so give an inflated LD? In other words, by having these minimum cutoffs are the authors getting an accurate estimation of LD for the region of the genome under study?

Individual genomic diversity

I understand why the authors looked at runs of homozygosity, but I don’t think it adds much to the story. I’d suggest removing to supplemental information.

Results

Gene annotation

Is there a link for interested researchers to download the annotated gene information?

Variant calling

Rewrite the first sentence. Maybe something like this “A total of 25,728,393 variants in 140 individuals were filtered to remove indels, SNPs with more than two alleles, maf <0.05, and were genotyped in more than 90% of samples to leave 4,533,143. These variants were further filtered to…”

Was there no limitation based on the number of reads necessary for scoring SNPs? Was there a minimum number of reads necessary for determining heterozygotes from homozygous genotypes? What did the authors do with homeologous regions?

Figure legends. This really confused me and I’d request that the authors move the figure legends to before the figures rather than embedding them in the results section.

Line 463: the wording is clumsy here. The nearness of the lowest p-value SNP to aldh9a1 is the important point.

Line 505: why do the authors think it’s unclear if cpa6 is the most likely gene under this association peak?

Line 555: is this one SNP or several? Where is it in terms of the genome, is it near any other genes?

Figures and Tables

Figure 1: again, please change this to a table. I think it would be easier to read and interpret.

Figure 2: the resolution is extremely bad and needs improving. The x and y axes in part A are confusing. Are these principle components 1 and 2? What are the two inlay figures? It is unclear from the figure legend. Parts B and C are interesting, but again it’s nearly impossible to read the text associated with the figures.

Figure 3: again, the resolution is poor and I’m not convinced the figure is necessary. There are a lot of figures that are showing, essentially, the same thing. I’d recommend removing figure 3 to the supplemental info.

Figure 4: part A: it’s difficult to read, but it seems that the Bonferroni significant cut off is above 15. How are the authors determining this? I thought Bonferroni significance is determined by dividing alpha by the sample size. In this case 0.05 / total number of SNPs (450868) should give a negative log10 P value of 6.955. I’m also confused about part C: The authors want to show how LD breaks down around the SNP with the strongest association but I’m struggling to see how LD breaks down in this figure. My suggestion would be to do a more traditional LD heatmap.

Figure 5: again, the quality is poor although the results are interesting. I think these data might be better presented in a table that reports on the comparison being made, i.e., Group 3 versus Group 2, the p-value, the gene, and the location of the gene on the sockeye genome.

Figure 6: The possibility of an inversion that appears to be associated with the groupings is very interesting. However, it’s difficult to interpret the figure and make the link between the data and how they support the hypothesis of an inversion. Would not a simple LD heatmap show the same data in an easier way? In addition, the possible inversion on chr 24 was not discussed. Are there other datasets that might point towards an inversion on chr24, or is this completely novel? If novel this needs to be better characterized.

Figure 7: part A seems unnecessary. Please tell the reader how many samples were included in this analysis and how many were kokanee versus anadromous sockeye. Part B is interesting but again I’m unsure how the lines for significance were drawn. Part C is interesting but needs clarity, what does the 1st, 2nd, 3rd, 4th refer to? I could not deduce that from the figure legend. I also think this is not the best way to present these data. A simple bar graph that shows the proportion of kokanee that are ancestral, heterozygous, and homozygous for the alternative allele and the same with anadromous sockeye would be far simpler.

Table 1: confusing and badly formatted. Please add gene abbreviations to a new column.

Figure 8: please format the axes in part E so that the distance is presented as MB rather than bases. I’d also recommend changing the scale bar for LD to make the distinction between areas with high LD and low LD clearer.

Figure 9: does not add to the paper, I’d suggest removing or moving to the supplemental information.

Discussion

The first sentence needs a rewrite. Put the contribution of the sockeye genome in the bigger context better. Something like “adds to a growing number of completed salmonid genomes”

Lines 601-603: what does this suggest?

Lines 609-610: take out the “with a slight discrepancy between…” it doesn’t add to the sentence.

Lines 613-614: what does this suggest with respect to the number of genetic populations and the potential isolation of samples from British Columbia?

Line 618: what does this suggest with respect to separation of kokanee and sockeye?

Lines 631-636: the separation of populations on the basis of immunoglobulin heavy chain is interesting and needs more exploration. Why might there be a difference at this locus? What about the gene duplication between chr21 and chr26 are these homeologous in sockeye?

Line 644: three loci have also been found to be associated with ecotype diversity in other studies? If so, how? Between kokanee and sockeye? Or between beach and stream spawning?

The whole section on genes associated with life history ecotype development needs work. For example, the authors mention neuregulin 3, but make no effort to discuss why that gene might be different between kokanee and sockeye. Same with the other genes connected to phototransduction, skeletal development, and immunity.

Line 674: what do the authors mean by conserved ecotype associations? Conserved between studies or between populations?

Line 691: why might aquaporin-3 alleles be associated with ecotype development?

Lines 694-695: is this different from how the Y chromosome formed in other salmonids? How do the homologs compare with Y chromosomes in other salmonids?

The lack of sdY in some populations of sockeye salmon is interesting and warrants further discussion.

Line 706: which chromosome is the kruppel-like factor 5 gene on?

---

## [Author Response · Author response to Decision Letter 0]

18 Aug 2020

Editor’s Comments:

 This was checked.

2. To comply with PLOS ONE submissions requirements, please provide methods of sacrifice in the Methods section of your manuscript.

 Added euthanasia protocols for samples taken for this study.

3. We note that you are reporting an analysis of a microarray, next-generation sequencing, or deep sequencing data set. PLOS requires that authors comply with field-specific standards for preparation, recording, and deposition of data in repositories appropriate to their field. Please upload these data to a stable, public repository (such as ArrayExpress, Gene Expression Omnibus (GEO), DNA Data Bank of Japan (DDBJ), NCBI GenBank, NCBI Sequence Read Archive, or EMBL Nucleotide Sequence Database (ENA)). In your revised cover letter, please provide the relevant accession numbers that may be used to access these data. For a full list of recommended repositories, see http://journals.plos.org/plosone/s/data-availability#loc-omics or http://journals.plos.org/plosone/s/data-availability#loc-sequencing.

 All raw sequences had already been submitted to NCBI’s SRA. The assembly already had an accession as well, but we added the NCBI BioProject ID so all 170 SRA accessions can be identified (e.g. https://www.ncbi.nlm.nih.gov/bioproject/PRJNA530256/) for the project.

4. In your Methods section, please provide additional location information of the sampling sites, including geographic coordinates for the data set if available.

 Added - Table with approximate geographic coordinates.

Reviewer #1 comments

1. Line 374-376, please add references.

 Added - NCBI annotation reports (NCBI website)

2. Line 372, the sockeye salmon, sockeye salmon,and coho salmon are closely related species. Why the protein-coding genes annotated in this species were much less than salmon and trout? Whether this indicate the poor quality of the genome assembled or some characteristics related to this species? Please discuss it in detail. In addition, what genes were not identified in this species?

 Added to the results section - gene count discrepancy between species, likely due to assembly quality differences and/or differences in RNA-seq data sets. Count discrepancies were found between version 1 and 2 of the coho salmon genome assemblies as well (version 1: 36,425 vs. version 2: 41,269) and likely represents quality differences. This is discussed further in the new results section (Orthology between species).

 Identifying missing genes between species is difficult as genes are inconsistently named between species with names like LOC123456789 in one species and the ortholog named LOC234567890 in another. This requires assignment of orthologs between species first.

 Added to the method section: Method for identifying orthologs between sockeye salmon and two other species. Supplemental figures show that “missing” genes are disproportionally found in the telomeric ends supporting that they are missing because of difficulties in assembling complex regions of the chromosomes (highly repetitive), which has been discussed previously (citation in text).

3. The authors identify the Chr.9a/9b as the sex chromosomes. However, no data was provided to support this conclusion because no marker was developed based on the differentiated regions on this chromosome to distinct the genetic male and female. The sockeye salmon has XX/XY or ZZ/ZW sex determination system? Please indicated it in the introduction.

 Chr 9a and 9b were identified as the X-chromosomes in a previous study. This is made more clear in the Introduction and more information is given about sex-determination in sockeye salmon. 

4. The authors analyze the genomic associations with kokanee ecotype. Some candidate genes, including aquaporin-3, trim45, etc, were identified and discussed in three populations. But no functional experiments were performed to demonstrate it. In addition, whether the expression of these genes showed different patterns in different populations?

 While it would be great to test these candidate genes, this manuscript’s scope was to identify populations and identify candidate genes that might be under selection. Future work will be to validate or eliminate these candidate genes as this work would take much longer to perform.

5. In Figure 2B, this figure was used in Figure 3, 4, 5 and 6. Please delete it.

 Figure 2B was removed and the text was updated based on this removal.

6. Species name in the References should be in italics.

 Changed.

Reviewer #2 comments

I thought the finding of the large region of divergence between lineages on chromosome 26 was very interesting. I’m wondering if this could be an inversion or if some other mechanism is contributing to the large peak. Maybe this could be similar to inversions between lineages in Atlantic salmon? (e.g. https://doi.org/10.1111/mec.15065) I recommend the authors add a few sentences discussing potential mechanisms that could explain this large peak.

 Added to the results section that this is possibly an inversion based on haplotypes. Added to the discussion that this may be a local adaptation that became fixed in glacial refugia and that is maintained between populations by reduced recombination and underdominance. Based on paired-end alignments, there wasn’t strong evidence that this was an inversion. This is discussed in the text and includes alternative mechanisms. 

Line 330: I think “covariants” should be “covariates”

 Changed to covariates (found twice in the manuscript and changed both).

Line 419: Hanzen should be Hansen

 Changed to Hansen.

Line 688: loci to locus

 Changed loci to locus in two locations in the manuscript.

Line 716: I thought the inclusion of the double haploid in the runs of homozygosity analysis was unnecessary. I recommended removing this individual from the analyses and taking the section describing these results out of the discussion.

 We agree that the doubled haploid result is obvious and that the manuscript would read better without this individual, but we think it is important to include. It isn’t completely clear from previous research that doubled haploids only retain the DNA from only one parent (please see article: Isogenic lines in fish – a critical review). This whole genome analysis of a doubled haploid might be useful for future comparisons and isogenic line development. However, we have reduced this discussion and it is only briefly mentioned now with most of the results and discussion removed.

Reviewer #3 comments

1) I feel that the manuscript suffers from an identity crisis. Is the main message to present the genome/protein coding parts of the genome or is the main aim of the manuscript to describe the analyses of the genetic data to answer the genetic basis of interesting questions? These needn’t be mutually exclusive, but I think the manuscript would read better if the authors could focus the writing on one of these two big picture aims.

 We have added a goal statement in the abstract to try to give the reader a better sense of the focus and goals. This should make the thesis of the manuscript more explicit. We have also tried to narrow the focus by moving several figures to supplemental data and removing unnecessary sentences guided by all of the reviewers’ comments.

2) The writing is poor and in places very difficult to read. Please give the manuscript a thorough re-read before resubmitting. Below in the minor comments I will detail some specific line numbers and sentences that were especially difficult for me to follow.

 Edits were made to simplify sentences and increase the ease of reading. Specific comments were addressed below.

3) The quality of the figures MUST be addressed. The DPI was too low to accurately read figures and this greatly diminished my enthusiasm for the manuscript. In addition, many of the figures seemed unnecessary and I would recommend moving to the supplemental data.

 The figures have gone through the PACE software for this journal and is at the recommended DPI. It may be that the figures need to be downloaded by the reviewer. Journals will often have low quality figures in auto-generated PDFs for review. They are meant to be downloaded separately. Figures 1, 5, 6, 7, and 9 were moved to supplementary.

4) Some analyses should be completed before resubmission. Specifically, I would like to see more information on how the authors dealt with the homeologous regions of the sockeye genome. Where they removed from this assembly? A circos plot, similar to previous salmonid genomes, would be very helpful in that regard.

 A circos plot was generated with common metrics shown. Homeologous regions are briefly addressed in the results section. The entire genome has homeologous regions and for the most part it appears they were successfully differentiated. It is difficult to both assemble and place contigs onto chromosomes for regions where the sequence similarity is still very high between homeologous regions. In lower-quality genome assemblies, these regions are typically left as contigs because there is not enough information to put them in the correct place on chromosomes. None of these regions were removed, but may have been left as unplaced contigs. Later versions of the genome assembly will be using long-read technology to address these issues.

Lines 28-30, arguably, this is a common feature of many different salmon and trout (e.g., rainbow trout and chinook). Maybe remove the “most complex and fascinating life histories” and instead say that “Repeatedly, a resident form known as kokanee…” and continue with the rest of lines 29-30.

 This line was removed.

Line 34: do a better job of linking these sentences, the polymorphisms within the immunoglobulin heavy chain are what’s causing a large part of the differentiation between the three groups.

 This sentence was altered to better link the two sentences and make it clear that it is the variants rather than the gene itself.

Paragraph starting on line 52 is confusingly written and much of the information seems unnecessary. The take home message is that sockeye exist as multiple different life history ecotypes and one of these is the freshwater resident kokanee. I’d recommend deleting this paragraph, taking the important message (i.e., what I say above) and adding this to the paragraph starting on line 61.

 Paragraph deleted and sentence added as requested.

Line 65: I’d suggest replacing “the hypothesis of…” with “is believe to be due to two common North American…”

 Changed.

Lines 69-71: this sentence must be clearer! The way I read it the kokanee appear to be monophyletic with respect to multiple rivers from the same area. Is that correct?

 That is correct. We have added a clarifying sentence.

Lines 75-76: selection with respect to what in Atlantic salmon? As in how is variation in this gene associated with selection?

 Added (associated with upstream catchment). 

Line 81: delete “assembly”.

 While it makes the sentence sound better if we remove assembly, it makes the statement inaccurate. The assembled sequence is a genome assembly and not a genome (for example: https://uswest.ensembl.org/Help/Faq?id=216). Instead, we removed “sequenced and assembled” and added generated in their place.

Lines 83-84: delete “various bodies of water”.

 Removed.

Lines 84-85: replace “that population structure” with ecotype divergence.

 Changed with a slight modification to let the reader know that we looked at divergence between populations and ecotype.

For the two fish used in genome sequencing and transcriptome sequencing please state what population these samples originated from. How old were the samples?

 These are Pitt Lake sockeye salmon (please see Samples section). The age was added to the Samples section.

Line 108/ figure 1: i’d recommend removing figure 1 as it is difficult to determine how many samples from each population there are and instead displaying that information as a table with the location, latitude and longitude, sample size, number of each sex, and the number of kokanee versus anadromous sockeye. This would be a useful resource for the reader as they progress reading the manuscript.

 Figure 1 was moved to supplemental material, but kept as many of these location would not be known to most people and the map gives an easier depiction of location than a table can. A table was added with aggregate information.

Lines 137-142: I am guessing that the samples were barcoded to allow pooling when sequenced? I’m assuming yes, but I can’t find the specific details in the methods either here or in the variant calling section. Some mention for how samples were barcoded and how sequences were separated by sample should be made.

 The NxSeq Adaptors were used to barcode samples from the NxSeq AmpFREE Low DNA Library Kit (please see text). All fastq files were recieved as single files (per individual) from McGill University and Génome Québec Innovation Centre. From my understanding, these files are generated with the standard Illumina software (please see: https://support.illumina.com/content/dam/illumina-support/documents/documentation/system_documentation/hiseqx/hiseq-x-system-guide-15050091-07.pdf). This is a standard practice if custom barcodes are not used.

Line 144: delete “that were sent”.

 Removed.

Line 147: the NEBNext RNA first strand synthesis is a way of synthesizing cDNA not for enriching extracted RNA for mRNA.

 This sentence has been modified to clarify.

Line 165: what quality filters were used?

 This is made more clear in the text (i.e. only adaptors were removed based on review of the output from FastQC)

Line 169-170: I don’t understand the “and using paired-end data” add on. I have a feeling this should be a separate sentence.

 Paired-end data (which has low error-rates) is used to correct the PacBio reads. This was made more clear in the manuscript.

Line 174: “corrected” PacBio reads, should that be filtered or quality filtered?

 PacBio reads are error prone and they are corrected—not filtered. For example: https://www.pacb.com/publications/lordec-accurate-and-efficient-long-read-error-correction/

Line 198: “found” in Christensen et al should be “described”.

 Changed.

Line 205: why were alignments filtered? What were the authors trying to remove?

 The following text was added to describe what was being filtered: “(e.g. off-target or repetitive elements).”

Line 210: please add sockeye to “previously published genetic maps”.

 Added.

Line 248: please replace “truth datasets” with to validate candidate SNPs.

 Truth set is the correct nomenclature and having that nomenclature is important for reproducibility. They were not used to validate candidate SNPs, they were used as a truth set in a model to score other variants. Please see: https://gatk.broadinstitute.org/hc/en-us/articles/360035890831-Known-variants-Training-resources-Truth-sets. 

Line 249: please replace “the truth without errors” with real.

 Changed.

Line 281-283: this sentence needs a rewrite. I’d recommend “Each of the methods used filtered variants to reduce the effects of high LD on subsequent analyses”.

 The sentence was rewritten: “To reduce the effects of high LD, variants that had been filtered for LD were used in the three clustering methods.”

Line 299-300: “from the clustering methodologies” (see clustering individuals section) should be deleted.

 Removed.

Lines 304-307: I’m confused by what was done here and why. What do the authors mean by “allele balance”? why wasn’t LD filtered?

 Reworded the sentence to simplify and added a mention to allele balance in the methods section for reference to what it means. Allele balance is the ratio between alleles. If for example you have the A/B genotype, you might have 100 A counts and 100 B counts, which is a 1:1 ratio (and expected based on chromosome segregation). If you have 10 A counts and 100 B counts that would suggest something is wrong with this locus. 

 For association studies, you typically have one phenotype that you are comparing at a time. In this case we are using LD1 values from the DAPC analysis for the phenotype. In order to tell what is specifically different between groups, only two groups could be compared at a time. 

 LD wasn’t filtered because that isn’t typically done and it would remove the peaks that are commonly found in association studies. For example, if we had a causal variant at 10,000 bp we would expect an association with this variant and other variants near it, but the p-values would increase as the distance increased from the causal variant due to recombination. If we filtered on LD, we would remove all but one of the variants in this region and possibly remove the causal variant or a variant that had a lower p-value than the one kept. 

Line 307: what was the p-value cutoff for the Bonferroni correction?

 Added that the alpha value was 0.01 before correction.

Lines 310-312: I am confused as to what data were used for this analysis and why.

 In this section, we were looking for chromosomal variation underlying population structure. There is an explanation of what EigenGWA is used for earlier in the paragraph. In the previous section, it was said that there were three grouping methods, DAPC, admixture, and a phylogenetic tree. In these lines (310-312), instead of the DAPC values used to identify groups, we used the admixture values to compare sockeye and kokanee from a subset of the samples. That was done because there was a difference in admixture values between kokanee and sockeye. This means there is a clear genetic difference between kokanee and sockeye in these geographic regions. By looking at this subset of individuals we are isolating that difference and by using the admixture values we are specifically looking for the part of the genome that best underlies this difference.

LD section

Please check the superscript formatting of R2 it seems off.

 This will be checked in preparation for publication. It looks good in the LibreOffice Writer text editor. 

I’m concerned about filtering for minimum R2 value around regions that might be in high LD. Would doing so give an inflated LD? In other words, by having these minimum cutoffs are the authors getting an accurate estimation of LD for the region of the genome under study?

 This was done for visualization and not used in any analyses. We were not trying to estimate LD in these regions only to show the markers that are associated with the population.

Individual genomic diversity

I understand why the authors looked at runs of homozygosity, but I don’t think it adds much to the story. I’d suggest removing to supplemental information.

 Figure 9 was moved to supplemental data. These metrics are an important aspect of sockeye salmon biology and the first attempt at finding them on this scale for the whole genome. They are also important for conservation and regulatory purposes and should therefore be reported to as wide of an audience as possible. Moving them to supplemental data would be counterproductive to this purpose.

Results

Gene annotation

Is there a link for interested researchers to download the annotated gene information?

 Added to references.

Variant calling

Rewrite the first sentence. Maybe something like this “A total of 25,728,393 variants in 140 individuals were filtered to remove indels, SNPs with more than two alleles, maf <0.05, and were genotyped in more than 90% of samples to leave 4,533,143. These variants were further filtered to…”

 Changed.

Was there no limitation based on the number of reads necessary for scoring SNPs? Was there a minimum number of reads necessary for determining heterozygotes from homozygous genotypes? 

 GATK works by generating a model of “good” variants that it then uses to score the rest of the variants. It may use read depth information, but the user does not supply a threshold and it wouldn’t be a set threshold in the results because other metrics might increase or decrease the score besides this value.

What did the authors do with homeologous regions?

 The entire genome is composed of homeologous regions, but for homeologous regions with very high sequence similarity we didn’t do anything different or special. What we did notice is that these regions tended to have much fewer variants. This could occur for a few reasons. The first reason is that reads would have a much lower mapping score because they could be mapped to multiple locations in the genome equally well. The GATK model would likely treat these regions as poorly supported and call fewer variants in them. Another reason could be that these regions have more recombination between homologous and homeologous chromosomes that could influence variant retention. After all, these regions have retained high sequence similarity > 95% (for regions that were able to be placed and this likely under represents this value) for around 90 million years. There must be a mechanism that reduces the accumulation of mutations in these regions to maintain that high sequence similarity for so long.

Figure legends. This really confused me and I’d request that the authors move the figure legends to before the figures rather than embedding them in the results section.

 This is a journal requirement.

Line 463: the wording is clumsy here. The nearness of the lowest p-value SNP to aldh9a1 is the important point.

 Changed to only say where the variant with the lowest p-value was found.

Line 505: why do the authors think it’s unclear if cpa6 is the most likely gene under this association peak?

 Added that the variant with the second lowest p-value is located in a nearby gene.

Line 555: is this one SNP or several? Where is it in terms of the genome, is it near any other genes?

 Added that this was multiple variants, and the approximate region and candidate gene. 

Figures and Tables

Figure 1: again, please change this to a table. I think it would be easier to read and interpret.

 Changed.

Figure 2: the resolution is extremely bad and needs improving. The x and y axes in part A are confusing. Are these principle components 1 and 2? What are the two inlay figures? It is unclear from the figure legend. Parts B and C are interesting, but again it’s nearly impossible to read the text associated with the figures.

 We believe the resolution issue is from the review manuscript quality generated by the journal. Journals often ask high-quality figures to be downloaded seperately for review (the same issue for all figures). Added what the axis are (linear discriminants, which are analogous to principle components). An error was found in this figure, with the DAPC groups mixed up on the DAPC figure. This was fixed throughout the manuscript.

Figure 3: again, the resolution is poor and I’m not convinced the figure is necessary. There are a lot of figures that are showing, essentially, the same thing. I’d recommend removing figure 3 to the supplemental info.

 Figure 1 was moved to supplemental data. This figure is the only one left for readers to be able to see where each group is relative to one another and is vital for understanding clustering. It would be very difficult to imagine this figure from latitude and longitude positions alone.

Figure 4: part A: it’s difficult to read, but it seems that the Bonferroni significant cut off is above 15. How are the authors determining this? I thought Bonferroni significance is determined by dividing alpha by the sample size. In this case 0.05 / total number of SNPs (450868) should give a negative log10 P value of 6.955. I’m also confused about part C: The authors want to show how LD breaks down around the SNP with the strongest association but I’m struggling to see how LD breaks down in this figure. My suggestion would be to do a more traditional LD heatmap.

 This was a coding error, log was used to draw the lines instead of log10. This was addressed in all figures and in the manuscript. This mistake made it necessary to edit several sections of the manuscript and to increase our criteria for a real peak. There were many more associations when considering the lower threshold. We increased the threshold from 3 nearby variants required to be considered a real peak to 5 to only analyze the most robust results. We thank the reviewer for catching this. It changed the results and discussion quite a bit. For part C, we were trying to show haploblocks and this is better explained in the manuscript now and shown in better detail in the figure.

Figure 5: again, the quality is poor although the results are interesting. I think these data might be better presented in a table that reports on the comparison being made, i.e., Group 3 versus Group 2, the p-value, the gene, and the location of the gene on the sockeye genome.

 This was changed to a table.

Figure 6: The possibility of an inversion that appears to be associated with the groupings is very interesting. However, it’s difficult to interpret the figure and make the link between the data and how they support the hypothesis of an inversion. Would not a simple LD heatmap show the same data in an easier way? In addition, the possible inversion on chr 24 was not discussed. Are there other datasets that might point towards an inversion on chr24, or is this completely novel? If novel this needs to be better characterized.

 A scatterplot with r2 values was added to the now supplemental figure to better visualize the block, in addition the possible inverted region(s) were highlighted. As far as we can tell, this possible inversion has not been mentioned in the literature before. We tried to confirm the potential inversion with paired-end data, but no convincing evidence was found. We are now using software to call inversions, but it may take much longer to figure out for sure and we believe this is outside the scope of this manuscript. For now we have discussed paired-end alignments and other mechanisms that might cause large haploblocks. 

Figure 7: part A seems unnecessary. Please tell the reader how many samples were included in this analysis and how many were kokanee versus anadromous sockeye. Part B is interesting but again I’m unsure how the lines for significance were drawn. Part C is interesting but needs clarity, what does the 1st, 2nd, 3rd, 4th refer to? I could not deduce that from the figure legend. I also think this is not the best way to present these data. A simple bar graph that shows the proportion of kokanee that are ancestral, heterozygous, and homozygous for the alternative allele and the same with anadromous sockeye would be far simpler.

 Part A was removed, and the sample number (sockeye salmon n=14, kokanee n=12) was added to both the methods section and the figure legend (now in supplementary). The threshold lines were redrawn with the correct values. For part C, the 1-4th lines referred to the significant variants with the lowest p-values. This is clarified in the text. We agree that a bar graph would be simpler, but we prefer this format as it allows the readers to see the haplotypes in this region for themselves. We reduced the size of the screenshot to simplify viewing.

Table 1: confusing and badly formatted. Please add gene abbreviations to a new column.

 This table was reformatted and gene abbreviations were added.

Figure 8: please format the axes in part E so that the distance is presented as MB rather than bases. I’d also recommend changing the scale bar for LD to make the distinction between areas with high LD and low LD clearer.

 Changed.

Figure 9: does not add to the paper, I’d suggest removing or moving to the supplemental information.

 Moved Figure 9 to supplementary figures. It is still discussed in the main text because these metrics are important for management, comparisons between species, and understanding the amount of genetic variation that can be commonly expected in this species.

The first sentence needs a rewrite. Put the contribution of the sockeye genome in the bigger context better. Something like “adds to a growing number of completed salmonid genomes”

 Changed.

Lines 601-603: what does this suggest?

 Added that is suggests that it is of lower quality.

Lines 609-610: take out the “with a slight discrepancy between…” it doesn’t add to the sentence.

 Removed.

Lines 613-614: what does this suggest with respect to the number of genetic populations and the potential isolation of samples from British Columbia?

 Added speculation as to why this might have occurred.

Line 618: what does this suggest with respect to separation of kokanee and sockeye?

 Added speculation as to why this might have occurred.

Lines 631-636: the separation of populations on the basis of immunoglobulin heavy chain is interesting and needs more exploration. Why might there be a difference at this locus? What about the gene duplication between chr21 and chr26 are these homeologous in sockeye?

 Added a possible explanation for why the immunoglobulin heavy chain might be involved. Added that chr21 and chr26 are homeologous.

Line 644: three loci have also been found to be associated with ecotype diversity in other studies? If so, how? Between kokanee and sockeye? Or between beach and stream spawning?

 Added to the discussion that the markers from the previous studies were aligned to the genome to enable comparison (genomic positions and marker names were already included in the discussion). Added the ecotype under comparison and analysis.

The whole section on genes associated with life history ecotype development needs work. For example, the authors mention neuregulin 3, but make no effort to discuss why that gene might be different between kokanee and sockeye. Same with the other genes connected to phototransduction, skeletal development, and immunity.

 Added discussion of why these genes might be connected to ecotype.

Line 674: what do the authors mean by conserved ecotype associations? Conserved between studies or between populations?

 Added to the sentence: (i.e. an association identified in the analysis between all sockeye and kokanee) 

Line 691: why might aquaporin-3 alleles be associated with ecotype development?

 This section was removed because aquaporin-3 did not pass the new threshold of 5 significant variants. 

Lines 694-695: is this different from how the Y chromosome formed in other salmonids? How do the homologs compare with Y chromosomes in other salmonids?

The lack of sdY in some populations of sockeye salmon is interesting and warrants further discussion.

 Added a section to the Introduction discussing salmonid Y chromosomes and comparisons. The lack of sdY is expounded on in the Discussion by adding details from other studies that have found sdY negative populations. The origin of the sdY gene is also discussed and how KLF5 might also influence sex.

Line 706: which chromosome is the kruppel-like factor 5 gene on?

 Added “(LG1 6,248,507 - 6256,452)” to the manuscript

---

## [Decision Letter · Decision Letter 1]

30 Sep 2020

PONE-D-20-12784R1

The sockeye salmon genome, transcriptome, and analyses identifying population defining regions of the genome

PLOS ONE

Dear Dr. Christensen,

Thank you for submitting your manuscript to PLOS ONE. After careful consideration, we feel that it has merit but does not fully meet PLOS ONE’s publication criteria as it currently stands. Therefore, we invite you to submit a revised version of the manuscript that addresses the points raised during the review process.

The manuscript has been reviewed by one of the previous referees. I agree with the referee that some minor revisions are still needed. I invite you to submit a revised version that address all the concerns arised. 

We look forward to receiving your revised manuscript.

Kind regards,

Zuogang Peng, Ph.D.

Academic Editor

PLOS ONE

Reviewers' comments:

Reviewer's Responses to Questions

**Comments to the Author**

1. If the authors have adequately addressed your comments raised in a previous round of review and you feel that this manuscript is now acceptable for publication, you may indicate that here to bypass the “Comments to the Author” section, enter your conflict of interest statement in the “Confidential to Editor” section, and submit your "Accept" recommendation.

Reviewer #3: All comments have been addressed

2. Is the manuscript technically sound, and do the data support the conclusions?

Reviewer #3: Yes

3. Has the statistical analysis been performed appropriately and rigorously? 

Reviewer #3: Yes

4. Have the authors made all data underlying the findings in their manuscript fully available?

Reviewer #3: Yes

5. Is the manuscript presented in an intelligible fashion and written in standard English?

Reviewer #3: Yes

6. Review Comments to the Author

Reviewer #3: Comments to the authors

The authors have addressed all of my comments and I feel the manuscript is now suitably for publication albeit with a couple of small changes. I would like to mention that I am impressed with the improvement in the quality of the writing and the authors should be commended in that regard.

Major comments

The only major comment I have concerns the length of the new version of the manuscript. I tried to find a section or two that could be moved to supplemental, but the only good candidate was the sampling details on lines 93-129 (as well as table one).

Minor comments

Lines 53-54: start the sentence with “This split between…” and then add “suggests” between “salmon species and” and “two common North American…”

Line 57: add “the phenotype is” to the text within brackets, “(i.e., the phenotypes are polyphyletic)”

I think adding some text to line 58 might help the reader understand why the Fraser and Colombia Rivers are different to the rest of the sockeye range. Perhaps something like “where multiple populations of Kokanee are more closely related to each other than sympatric sockeye salmon”.

Line 140: add how large the size selected fragment of DNA was.

Line 174: add the parameters used for filtering with FastQC.

Paragraph titled “clustering and chromosomal variation underlying population structure

This paragraph is very interesting, but the important points get a bit lost as the reader tries to keep which population is which straight. I’d suggest re-writing this section perhaps to emphasize that what’s interesting here is the discrepancy in the normal dogma that sympatric kokanee and sockeye are more closely related to each other than either is to allopatric populations of the same ecotype. Perhaps, the authors should start by saying “There are three clusters, one of which was composed of samples from the upper Columbia” and then go on to talk about the discrepancy between studies with regard to some kokanee forming a monophyletic group.

Lines 750-753: the information in brackets could be deleted as the methods used are well explained in the methods section.

7. PLOS authors have the option to publish the peer review history of their article (what does this mean?). If published, this will include your full peer review and any attached files.

Reviewer #3: No

---

## [Author Response · Author response to Decision Letter 1]

30 Sep 2020

Comments from reviewer 3

Major comments

The only major comment I have concerns the length of the new version of the manuscript. I tried to find a section or two that could be moved to supplemental, but the only good candidate was the sampling details on lines 93-129 (as well as table one).

 The sample section was moved to S1 Methods.

Minor comments

Lines 53-54: start the sentence with “This split between…” and then add “suggests” between “salmon species and” and “two common North American…”

 This was changed.

Line 57: add “the phenotype is” to the text within brackets, “(i.e., the phenotypes are polyphyletic)”

I think adding some text to line 58 might help the reader understand why the Fraser and Colombia Rivers are different to the rest of the sockeye range. Perhaps something like “where multiple populations of Kokanee are more closely related to each other than sympatric sockeye salmon”.

 This was changed.

Line 140: add how large the size selected fragment of DNA was.

 Added that the peak was around 488 bp.

Line 174: add the parameters used for filtering with FastQC.

 Added that default settings were used.

Paragraph titled “clustering and chromosomal variation underlying population structure

This paragraph is very interesting, but the important points get a bit lost as the reader tries to keep which population is which straight. I’d suggest re-writing this section perhaps to emphasize that what’s interesting here is the discrepancy in the normal dogma that sympatric kokanee and sockeye are more closely related to each other than either is to allopatric populations of the same ecotype. Perhaps, the authors should start by saying “There are three clusters, one of which was composed of samples from the upper Columbia” and then go on to talk about the discrepancy between studies with regard to some kokanee forming a monophyletic group.

 Changed to make this result more clear.

Lines 750-753: the information in brackets could be deleted as the methods used are well explained in the methods section.

 Removed.

---

## [Editor Report · Decision Letter 2]

6 Oct 2020

The sockeye salmon genome, transcriptome, and analyses identifying population defining regions of the genome

PONE-D-20-12784R2

Dear Dr. Christensen,

We’re pleased to inform you that your manuscript has been judged scientifically suitable for publication and will be formally accepted for publication once it meets all outstanding technical requirements.

Kind regards,

Zuogang Peng, Ph.D.

Academic Editor

PLOS ONE
---

## [Editor Report · Acceptance letter]

9 Oct 2020

PONE-D-20-12784R2 

The sockeye salmon genome, transcriptome, and analyses identifying population defining regions of the genome 

Dear Dr. Christensen:

I'm pleased to inform you that your manuscript has been deemed suitable for publication in PLOS ONE. Congratulations! Your manuscript is now with our production department. 

Kind regards, 

on behalf of

Dr. Zuogang Peng 

Academic Editor

PLOS ONE